# Kdm1a safeguards the topological boundaries of PRC2-repressed genes and prevents aging-related euchromatinization in neurons

Beatriz del Blanco [1,7,8] ✉, Sergio Niñerola [1,7], Ana M. Martín-González [1,7], Juan Paraíso-Luna [1,2], Minji Kim[3,4], Rafael Muñoz-Viana[1,5], Carina Racovac[1], Jose V. Sanchez-Mut [1], Yijun Ruan[3,6] & Ángel Barco [1,8] ✉

Kdm1a is a histone demethylase linked to intellectual disability with essential roles during gastrulation and the terminal differentiation of specialized cell types, including neurons, that remains highly expressed in the adult brain. To explore Kdm1a's function in adult neurons, we develop inducible and forebrain-restricted Kdm1a knockouts. By applying multi-omic transcriptome, epigenome and chromatin conformation data, combined with super-resolution microscopy, we find that Kdm1a elimination causes the neuronal activation of nonneuronal genes that are silenced by the polycomb repressor complex and interspersed with active genes. Functional assays demonstrate that the N-terminus of Kdm1a contains an intrinsically disordered region that is essential to segregate Kdm1a-repressed genes from the neighboring active chromatin environment. Finally, we show that the segregation of Kdm1a-target genes is weakened in neurons during natural aging, underscoring the role of Kdm1a safeguarding neuronal genome organization and gene silencing throughout life.

Each of the cell types in a multicellular organism express a different set of genes and accomplishes distinct functions, yet they share the same genetic material. This is achieved through epigenetic mechanisms that regulate the accessibility and packaging of the genetic material inside the cell nucleus. Chromatin has been classically classified into euchromatin and heterochromatin, which are linked to active transcription and gene silencing, respectively[1]. On a smaller scale, the genome is organized in topologically associating domains (TADs)[2,3] in which extensive chromatin interactions are observed. The basic units of chromatin interactions are chromatin loops involving two looping anchors[4,5] and are mediated by specific protein factors[6]. It is further understood that many chromatin architectural proteins, such as the CCCTC-binding protein (CTCF)[5,7], are critical for the establishment and maintenance of the overall chromatin folding architectures, which provide a genome-wide framework for gene expression that underlie cell identity. Liquid-liquid phase separation (LLPS) and related phase transitions recently emerged as a biophysical mechanism for the assembly of compartmentalization in distinct liquid-like structures

[1]Instituto de Neurociencias (Universidad Miguel Hernández - Consejo Superior de Investigaciones Científicas). Av. Santiago Ramón y Cajal s/n. Sant Joan d'Alacant, 03550 Alicante, Spain. [2]Universidad Complutense de Madrid, 28040 Madrid, Spain. [3]The Jackson laboratory for Genomic Medicine, Farmington, CT 06030, USA. [4]Present address: Department of Computational Medicine and Bioinformatics, University of Michigan, Ann Arbor, MI 48109, USA. [5]Present address: Bioinformatics Unit, Hospital universitario Puerta de Hierro Majadahonda, 28220 Majadahonda, Spain. [6]Present address: Life Sciences Institute, Zhejiang University, Hangzhou, Zhejiang Province 310058, P.R. China. [7]These authors contributed equally: Beatriz del Blanco, Sergio Niñerola, Ana M. Martín-González. [8]These authors jointly supervised this work: Beatriz del Blanco, Ángel Barco. ✉e-mail: bblanco@umh.es; abarco@umh.es

that participate in transcriptional control and other chromatin-related functions of nuclei[8].

Among the candidate molecules to regulate chromatin compartmentalization, we focused our attention on the lysine (K)-specific demethylase 1 A (Kdm1a; also known as LSD1). This flavin-dependent monoamine oxidase is linked to the action of different repressor complexes, such as coREST[9] and NURD[10], where it primarily promotes repression via H3K4me1/me2 demethylation. However, Kdm1a may also promote transcriptional activation at some loci in a context-dependent manner[11–13]. Although seminal genetic studies in fission yeast[14] and flies[15] implicated Kdm1a homologs in the regulation of boundaries between silenced and active chromatin domains, this role has not been confirmed in mammalian cells nor investigated using epigenomics methods.

The investigation of Kdm1a conventional knockout (KOs) mouse strains has demonstrated that Kdm1a is required for gastrulation and its germinal elimination is embryonic lethal[16,17]. Subsequent studies in conditional or tissue-restricted KOs have shown that Kdm1a is also important at later developmental stages for the differentiation and survival of several cell types, including neurons[18–21]. Furthermore, *Kdm1a* mutations have been linked with a rare monogenic intellectual disability disorder (IDD) known as Cleft palate, psychomotor retardation, distinctive facial features, and intellectual disabilities (CPRF; OMIM#616728)[22,23]. The discovery of a neuronal specific dominant-negative splicing isoform known as neuroLSD1 has further underscored the role of this enzyme in the nervous system[24]. However, its functional role in mature neurons of the adult brain remains unknown.

Here, we investigate an inducible mouse strain wherein Kdm1a is specifically eliminated in principal neurons of the forebrain during adulthood. Through a combination of multi-omic analysis, super-resolution microscopy, and functional assays, we provide insight into the mechanism of gene repression by Kdm1a in adult neurons and unveil the functions of this protein orchestrating chromatin compartmentalization and promoting nuclear phase separation.

## Results

### Efficient and cell type specific Kdm1a deletion in adult forebrain neurons

To investigate the role of Kdm1a in postmitotic, highly specialized cells, we focused on excitatory neurons in the adult brain. We generated tamoxifen (TMX)-regulated, forebrain-restricted *Kdm1a* KOs (referred to as Kdm1a-ifKO) by crossing *Kdm1a* ^f/f^ and *Camk2α*-creER^T2^ mice (Fig. 1A). After treatment with TMX at 2 months of age, Kdm1a-ifKOs showed an efficient and specific loss of Kdm1a in principal neurons of the forebrain, including pyramidal neurons in CA1, CA3, and cortex, and granular neurons in the dentate gyrus (Fig. 1B, C and Supplementary Fig. 1A, B), whereas other brain areas and cell types in which the *Camk2α* driver is not active, such as the cerebellum and astrocytes, presented unaltered Kdm1a expression (Supplementary Fig. 1A–E). Kdm1a-ifKOs displayed normal weight and growth (Supplementary Fig. 1F). Neither DAPI counterstaining nor immunostaining with neuronal markers revealed any changes in the neuronal nuclei, shrinkage, or other histological alterations in the hippocampus (Supplementary Fig. 1G, H). Moreover, Kdm1a-depleted neurons did not show pyknosis, expression of active Caspase-3 (Supplementary Fig. 1H), nor were positive in a TUNEL cell death assay (Supplementary Fig. 1I). We did not detect either the expression of inflammation or cell death markers (Supplementary Fig. 1J). Further supporting the viability of the neurons lacking Kdm1a, these cells still showed a normal response to stimulation even more than one year after *Kdm1a* ablation (Supplementary Fig. 1K, L). These results contrast with the widespread neuronal loss in hippocampus and cortex, and premature death displayed by the conditional KOs reported by[21]. Since that article described a strain with inducible (TMX-regulated), but ubiquitous (no neuronal-specific) elimination of *Kdm1a*, the two

studies together indicate that although *Kdm1a* is not needed cell autonomously for the viability of principal neurons, it is likely required in other cell types that support neuronal survival. Consistent with this view, and in agreement with information found in public repositories[25] (Supplementary Fig. 1M), we detected *Kdm1a* expression in support brain cells types such as astroglia and endothelial cells (Supplementary Fig. 1C, D).

### Ablation of Kdm1a in adult excitatory neurons derepresses nonneuronal, PRC2-repressed genes

We next investigated whether the elimination of *Kdm1a* in adult neurons triggered transcriptional changes. The bulk RNA-seq analysis of hippocampal tissue, combined with previous nuRNA-seq profiling of hippocampal excitatory neurons[26], uncovered that Kdm1a is the highest expressed H3K4 KDM in adult hippocampal neurons (Fig. 1D). The analysis also demonstrated the absence of compensatory upregulation of other KDM-encoding genes or the downregulation of lysine methyltransferases (KMT) -encoding genes, at least at the transcriptional level. The RNA-seq screen comparing Kdm1a-ifKOs and control littermates confirmed that Kdm1a ablation did not induce inflammation nor the expression of cell death markers. The great majority of the differentially expressed genes (DEGs) in the hippocampus of Kdm1a-ifKOs were upregulated (252 out of 267, > 94%; p adj <0.05; |log$_2$ Fold Change (FC)| > 1; Fig. 1E and Supplementary Data 1) and showed weak or no expression in the hippocampus of control mice (Fig. 1F and Supplementary Fig. 2A). These results indicate that, although Kdm1a neuronal function has been described as bivalent, acting both as transcriptional repressor and activator in developmental processes and cell lines depending on the associated protein complexes[27], in mature neurons Kdm1a primarily acts as a repressor. Intriguingly, Gene Ontology (GO) enrichment analysis of the upregulated DEGs (upDEGs) retrieved *Biological process* terms not related to neurons, such as *Peptidyl-lysine oxidation*, *Hepatocyte differentiation*, *Epithelium development* or *Regulation of insulin secretion* (Fig. 1G). There are many examples of upDEGs whose expression is normally restricted to other tissues than the brain. For instance, *Bmp8a* and *Spint1*, two of the most significantly upregulated genes in Kdm1a-ifKO mice, are expressed in endocrine tissues and the gastrointestinal tract, respectively, but not in neurons (NCBI database, www.ncbi.nlm.nih.gov). RNA-seq analysis also unveiled the upregulation of genes that are only expressed in neurons during the early stages of development and become silent in mature neurons. This is the case of *Prph* (peripherin), a component of the cytoskeleton that during adulthood is only expressed in neurons of the peripheral nervous system[28]. The diversity of GO terms associated with upDEGs suggests a common repressive mechanism for different nonneuronal genes.

To further explore the regulatory circuitry regulated by Kdm1a we analyzed transcription factor (TF) binding enrichment in DEGs. The proteins showing the most significant enrichment in upregulated genes were Suz12 and Ezh2 (Fig. 1H), two essential subunits of the Polycomb Repressive Complex 2 (PRC2) that preferentially bind to cell type-specific repressed gene promoters[29]. Consistent with this result, the transcription start site (TSS) of upDEGs show low accessibility in neuronal-specific ATAC-seq in hippocampal neurons from wild type mice, concomitantly with the absence of histone posttranscriptional modification (hPTM) associated with active transcription (e.g., H3K4me3, H3K27ac), lack of transcriptional co-activators (e.g., CBP) and RNA polymerase II (RNAPII) binding, and a prominent presence of the hPTM H3K27me3 (Fig. 1I). The trimethylation of H3K27 is catalyzed by the PRC2 and numerous publications demonstrate that H3K27me3-rich genomic regions function as silencers to repress gene expression via chromatin interactions[30–32]. H2AK119ub1, a hPTM that stabilizes canonical PRC1 and PRC2 activities[33], was also highly enriched at the TSS of upDEGs both in the chromatin of neuroprogenitor cells (NPCs) and the postnatal mouse brain (mBrain) (Supplementary Fig. 2B-C).

 

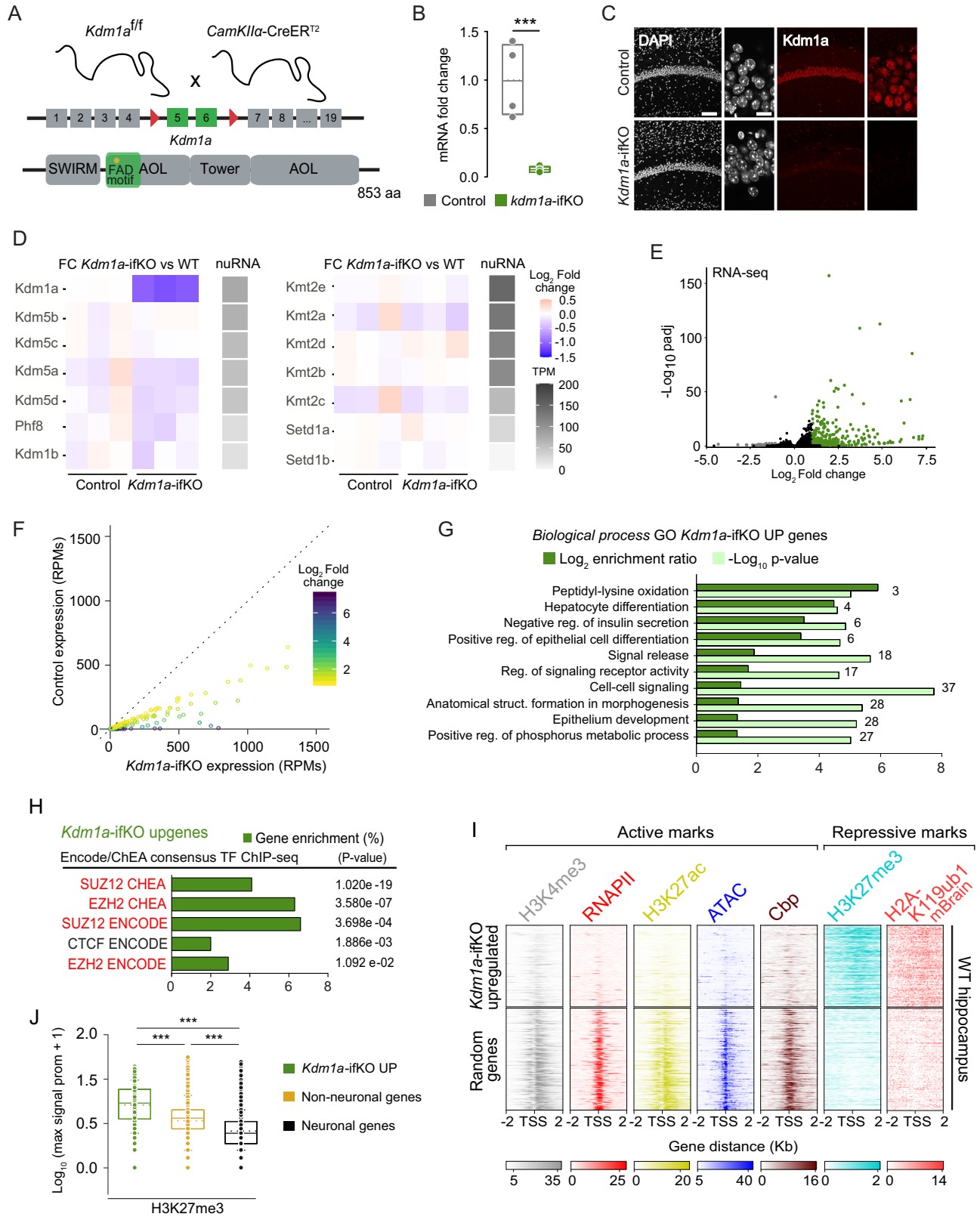

The largest deposition of H3K27me3 in Kdm1a-dependent genes, compared to both, other nonneuronal genes and genes expressed in neurons (Fig. 1J), underscores a cooperative role between Kdm1a and PRC2 in the repression in neurons of genes that are specifically expressed in other cell types. Although previous studies in non-neuronal cells described a physical and/or functional interaction of Kdm1a and the PRC2 complex subunits[34,35], these analyses indicate that such interaction also occurs in mature neurons in the adult brain. Note that the mediator of this interaction in nonneuronal cells, the long non-coding RNA *Hotair*[35], is not expressed in mature excitatory neurons (Supplementary Fig. 2D); therefore, alternative mechanisms must be in place.

**Fig. 1 | Kdm1a loss in adult excitatory neurons causes de-repression of PRC2-repressed nonneuronal genes. A** Genetic strategy used to deplete *Kdm1a*. Floxed exons and the encoded flavin adenine dinucleotide (FAD)-binding domain are highlighted. Kdm1a also has amine oxidase-like (AOL), SWIRM and tower domains. **B** *Kdm1a* transcript levels 4 weeks after TMX administration ($n = 6$ Kdm1a-ifKO; $n = 4$ control (CT); T = 34, $p = 0.01$, Mann-Whitney test, two-tailed). Box plots indicate median value, interquartile range, minimum and maximum value (whiskers), and individual data points. **C** CA1 immunostaining with anti-Kdm1a 2 months after TMX. DNA was counterstained with DAPI. Scale bars: 100 μM and 10 μm in insets. **D** Gray scale: Nuclear RNA levels for KDMs and KTMs in excitatory forebrain neurons[26] in transcripts per million (TPM). Color scale: Log$_2$FC between genotypes. **E** Volcano plot of differential expression analysis ($n = 3$ CT and 3 Kdm1a-ifKO). Upregulated genes are labeled in green and downregulated genes in grey (Wald test, p adj <0.05; |log$_2$ FC | > 1). **F** Scatter plot of upDEGs expression in Kdm1a-ifKOs and controls. **G** GO enrichment analysis for *Biological processes* in upDEGs (Fisher's exact test). Number of genes, enrichment ratio and statistical significance are shown. **H** Enrichment analysis for TF binding in upDEGs (Fisher's exact test, p adj <0.05; log$_2$FC > 1). PRC2 subunits are highlighted in red. **I** Density of ChIP-seq reads for H3K4me3, H3K27ac, accessibility measured by ATAC-seq, binding of CBP and RNAPII, and the PRC2-related repressive marks H3K27me3 and H2AK119ub. Heat-maps represent the enrichment for all these marks in hippocampal chromatin of wild type (WT) mice around the TSS (± 2 Kb) of upDEGs and the TSS of a similarly sized set of random genes expressed in neurons. **J** Quantification of H3K27me3 signal in the set of upDEGs ($n = 514$) compared with neuronal genes ($n = 14902$) and with nonneuronal genes not affected by Kdm1a loss ($n = 11372$; Mann–Whitney U Statistic, two tailed, pval ***<0.0001). Box plot as described in panel B except for whiskers showing SD values. Source data are provided as a Source data file.

## Gene derepression is accompanied by dramatic changes in epigenetic profiles

Since *Kdm1a* has a direct impact on the methylation status of H3K4, we next performed chromatin immunoprecipitation coupled with high-throughput sequencing (ChIP-seq) for H3K4me3 and H3K4me1 in the hippocampi of adult Kdm1a-ifKOs and control littermates. H3K4 differentially methylated regions (DMRs) showed an increase of H3K4me3 levels (Fig. 2A; FDR < 0.1: 357 regions), the majority of which overlapped with promoter regions (Supplementary Fig. 3A). Consistent with the association of this hPTM with active transcription, we detected a significant increase of H3K4me3 in the promoter of upDEGs in line with the magnitude of the transcriptional upregulation (Fig. 2B, C and Supplementary Fig. 3B, C). Furthermore, changes in transcript and H3K4me3 levels showed a strong correlation (Supplementary Fig. 3D). In parallel to H3K4me3 increase we observed an apparent depletion of H3K4me1 at the same TSSs (Fig. 2B), confirming the unbalance of H3K4 methylation caused by Kdm1a loss. Since the loss of Kdm1a resulted in increased H3K4 methylation at enhancers in mouse embryonic stem cells[36], we also looked for changes in enhancer regions but did not observe differences between genotypes (Supplementary Fig. 3E).

Similar to what was observed for DEGs in Fig. 1H, the factors showing the most significant binding to DMRs in Kdm1a-ifKOs neurons were Suz12 and Ezh2 (Fig. 2D), indicating once more functional interaction between Kdm1a and PRC2. Furthermore, also similar to DEGs (Fig. 1I), DMRs displayed low levels of H3K4me3 and other active transcription marks, alongside high levels of H3K27me3 and H2AK119ub1 within the neuronal chromatin of control mice (Fig. 2E). Since the increase of H3K4 methylation induced by the loss of Kdm1a may cause the disengagement of PRC2 and consequently decrease H3K27 methylation disrupting the positive feedback loop that underlie the repression of transcription at targeted regions[37], we next assessed whether Kdm1a loss affected the levels of H3K27me3 at upDEGs. ChIP assays revealed a strong reduction of H3K27me3 at promoter regions (Fig. 2F) that was accompanied by an increase in H3K27ac in the same upDEGs (Fig. 2G). To examine these changes genome-wide, we performed ChIP-seq using antibodies against the same hPTMs. We found that Kdm1a-DEGs displayed a reduction of H3K27me3 levels that were consistently associated with an increase in H3K27ac signal in the same genes (Fig. 2H, I and Supplementary Fig. 3F). These changes could not be attributed to the downregulation of PRC1 or PRC2 because the mRNA and protein levels of PRC subunits were similar in Kdm1a-ifKOs and control littermates (Supplementary Fig. 3G, H), and no global reduction of H3K27me3 was observed (Supplementary Fig. 3I). For instance, many classical PRC targets, such as the clusters of *Hox* genes, were unaffected by Kdm1a loss (Supplementary Fig. 3J).

## Topological association between upDEGs

Interestingly, the analysis of the genomic distribution of upDEGs revealed significant enrichment in specific chromosomal domains and a non-random distribution throughout the genome (Fig. 3A, B). Kdm1a's target genes tended to be in closer proximity than would be expected by chance and were often found in pairs or larger clusters. We next took advantage of our recent Hi-C analysis in hippocampal principal neurons[26] to analyze the distribution of DEGs between chromatin compartments A and B, which strongly correlate with euchromatin and heterochromatin, respectively[4]. Regions inside each compartment tend to interact with each other, while interactions between regions in different compartments are virtually absent[38,39]. As expected, most of the genes that are expressed in adult neurons are located into the A compartment (Fig. 3C). Notably, most upDEGs (92%), despite their lack of expression in wild-type neurons, were mapped into the A compartment interspersed with active genes (Fig. 3C, D and Supplementary Fig. 4A) and exhibited a frequency of DNA-DNA interaction with active domains similar to neuronal genes (Fig. 3E). We did not observe a differential distribution of upDEGs between the A1 and A2 sub-compartments defined in[40] (Supplementary Fig. 4B). Intriguingly, the loss of Kdm1a might also prevent the repression of neuronal genes interspersed in a repressive environment because 60% of the genes downregulated in *Kdm1a*-iKOs located into the B compartment despite being expressed in wild type neurons (Fig. 3C and Supplementary Fig. 4C, D). Only 15 genes were downregulated (p adj <0.05; |log$_2$FC| > 1), and they were not further analyzed.

Since CTCF motifs are highly enriched at the regulatory regions of upDEGs (Fig. 1H), we hypothesized that Kdm1a-sensitive genes might be connected through CTCF interactions[41]. Supporting this view, the elimination of PRC2 in postmitotic neurons, similar to Kdm1a, triggers the expression of nonneuronal genes[42]. Furthermore, many upDEGs directly bind CTCF[43] (Supplementary Fig. 4E, F), and a differential expression screen in peripheral neurons of CTCF conditional KOs[44] retrieved transcriptional changes that overlapped with upDEGs (Supplementary Fig. 4G). To test the hypothesis that Kdm1a-sensitive genes associate through CTCF looping, we analyzed CTCF-dependent interactions in excitatory neurons in the adult hippocampus using a reporter strain that allows the tagging and sorting of nuclei from excitatory hippocampal neurons[26] (Supplementary Fig. 4H). Sorted nuclei were used for Chromatin Interaction Analysis by Paired-End Tag Sequencing (ChIA-PET), a technique that combines chromatin proximity ligation with ChIP-based enrichment[45] (Fig. 3F). The ChIA-PET analysis (Supplementary Data 2) unveiled the interaction among upDEGs situated within the same chromosomal domains, suggesting the potential for proximate upDEGs to be physically brought closer together (in a 3D space) through CTCF interactions (Fig. 3G–G´). Similar results were observed when we crossed the genomic location of DEGs with Hi-C interaction maps for hippocampal excitatory neurons in the adult mouse brain[26] and in embryonic cortical neurons[46] (Supplementary Fig. 4I–K). ChIA-PET data also revealed that Kdm1a-sensitive genes show a stronger association with CTCF and locate, on

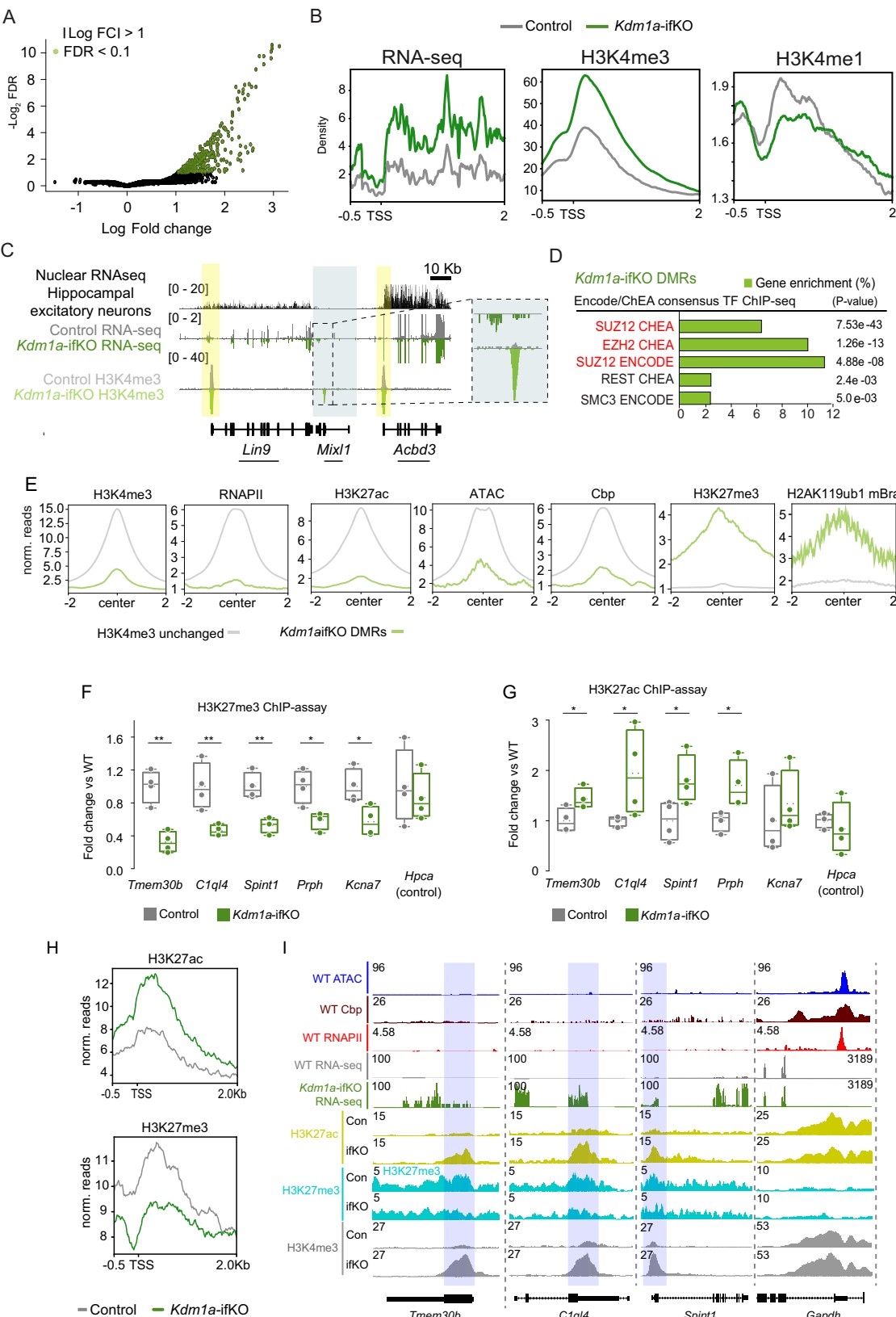

average, in smaller loops than both neuronal genes and nonneuronal genes not affected by Kdm1a loss (Fig. 3H). These findings suggest that Kdm1a loss preferentially affects PRC2-repressed genes that are clustered into an active genomic environment and are subjected to tight topological control.

**Topological association between Kdm1a and its target genes**

To explore the relationship between transcriptional and epigenetic changes and Kdm1a binding in neurons, we next conducted a ChIP-seq experiment using Kdm1a antibodies. The profile for Kdm1a binding in hippocampal chromatin of adult mice was very similar to that in adult

**Fig. 2 | Gene de-repression is accompanied by dramatic changes in the epigenetic profile. A** Volcano plot of H3K4me3 DMRs ($n = 2$ for CT; $n = 3$ for Kdm1a-ifKO; FDR < 0.1, |log₂FC | > 1). **B** Density distribution of RNA-seq, H3K4me3, and H3K4me1 reads around the TSS of upDEGs (distance for TSS, −0.5 Kb, and +2 Kb). **C** Genomic snapshot around *Mixl1*, as an example of upregulated gene. The graph shows mRNA-seq and H3K4me3 profiles in the adult hippocampus of Kdm1a-ifKO and control littermates. The upper nuclear RNA-seq track shows that *Mixl1* is not expressed in excitatory forebrain neurons[26], whereas neighboring genes are highly expressed. Data range is shown in brackets. Arrows indicate the direction of transcription. **D** Enrichment analysis for TF binding at H3K4-DMRs in Kdm1a-ifKOs. Subunits of PRC2 are highlighted in red (Fisher exact test, p adj <0.05; log₂FC > 1). **E** Graphs show the enrichment for active and repressive marks around the center (± 2 Kb) of H3K4-DMRs (green track; FDR < 0.1) compared to regions without

changes in H3K4me3 (grey track; FDR > 0.1) in the chromatin of WT mice. **F, G** H3K27me3 (**F**) and H3K27ac (**G**) ChIP-assays in Kdm1a-ifKO and control littermates revealed changes in several upDEGs (CT, $n = 4$; Kdm1a-ifKO, $n = 4$). The *Hpca* gene is shown as a negative control. Boxplots indicate median value, interquartile range, minimum and maximum value, and individual data points (two-tailed t-test, **$p$-val < 0.001; *$p$-va12 < 0.01). **H** Density distribution of H3K27ac and H3K27me3 reads around the TSS of upDEGs (distance to TSS, −0.5 Kb, and +2 Kb) in Kdm1a-ifKOs and control littermates. Compare with the H3K4me3 and RNA profiles presented in panel 2B. **I** Genomic snapshots show three examples of upDEGs, *Tmem30b, C1ql4,* and *Spint1*; and one non-changing gene, *Gapdh*. Chromatin accessibility, CBP, and RNAPII ChIP-seq profiles from hippocampus of adult mice are plotted. Source data are provided as a Source data file.

mouse prefrontal cortex (PFC)[47] (Supplementary Fig. 5A). Kdm1a binding was observed at promoters and enhancers with high levels of H3K4me3, RNAPII, H3K27ac, CBP and accessibility and low levels of the repressive mark H3K27me3 (Fig. 4A and Supplementary Fig. 5B). This pattern sharply differs from that found in Kdm1a-ifKO upDEGs (Fig. 1I). Further analyses indicated that Kdm1a resides in the A compartment together with the genes deregulated upon its removal (Fig. 4B) although it does not directly bind to most of them. Only a relatively small percentage of the most upregulated genes showed prominent Kdm1a binding in the ChIP-seq analysis (log₂FC > 1: 27 genes; 10.7%; Supplementary Fig. 5C, D). These results raised the question of how an epigenetic repressor that binds to active genes exerts its repressive function on targets that are far-off in a linear genome.

One possible explanation would be that topological interactions bring Kdm1a to the proximity of its targets (Fig. 4C). Supporting this hypothesis, some CTCF loops presented H3K27me3 at one CTCF anchor and H3K27ac and Kdm1a at the opposite one (Supplementary Fig. 6A). We observed that many upDEGs were involved in CTCF loops that approached the H3K27me3-rich region at the gene body of upDEGs to H3K27ac-rich regions occupied by Kdm1a (Supplementary Fig. 6B, C). To explore in greater detail this pattern, we plotted the metagene profiles for both the upDEGs and the genes connected to the other side of the CTCF loop. The metagene graphs showed that the increase in transcript level for upDEGs was not accompanied by transcriptional changes in the highly transcribed associated genes. However, in the case of H3K27ac we observed an apparent transfer of this mark between the active gene and the upDEGs, in parallel to an increase of H3K4me3 and a reduction of H3K27me3 in the chromatin of Kdm1a-ifKO mice (Fig. 4D). The snapview for the interaction between *Spint1* and two highly transcribed genes, *Vps18* and *Ino80*, that are topologically connected to *Spint1* according to the ChIA-PET data, illustrates this effect (Fig. 4E; Supplementary Fig. 6D shows another example loop).

We next investigated whether the absence of Kdm1a affected the interactions of the *Spint1* promoter. To assess this, we used chromosome conformation capture sequencing (4C-seq) to examine the short and long-range chromatin interactions of *Spint1*, a representative upregulated gene (Supplementary Fig. 6E). Comparing the 4 C results between Kdm1a-ifKO and control mice, we observed a decrease in the frequency of interactions between the *Spint1* and *Ino80*, along with higher interaction frequencies with other genes that are also upregulated in Kdm1a-ifKO neurons (Fig. 4F, G, Supplementary Fig. 6F–H and Supplementary Data 3). These results suggest that upon Kdm1a's removal, the boundaries of the three-dimensional structure that encompasses Kdm1a-sensitive genes weaken, allowing for the spreading of active marks into the repressed genes.

### Kdm1a is involved in chromatin compartmentalization
To clarify the mechanisms by which Kdm1a contributes to the maintenance of chromatin boundaries in a 3D space, we first tested whether Kdm1a loss destabilized CTCF loops. ChIP experiments

in hippocampal chromatin of Kdm1a-ifKOs and control littermates demonstrated that the binding of CTCF to upregulated genes was not affected by the loss of Kdm1a (Supplementary Fig. 6I). This result indicates that although Kdm1a repressor activity may depend on CTCF-mediated chromatin folding, its loss did not affect CTCF binding.

Next, we tested whether the distribution of active and inactive compartmentation was perturbed in the nucleus of Kdm1a-ifKO neurons. We used super-resolution microscopy to explore the spatial distribution of Kdm1a, H3K27me3, and H3K27ac inside the nucleus of hippocampal neurons. In agreement with ChIA-PET results, the images revealed the presence of Kdm1a in the proximity of some H3K27me3 puncta (Fig. 5A; separation between the centroids of Kdm1a and H3K27me3 puncta: $86 \pm 27$ nm). In control mice, H3K27me3 puncta were predominantly associated with highly condensed chromatin structures[48], such as chromocenters and peripheral heterochromatin, while excluding the core region of chromocenters. In contrast, H3K27me3 puncta were more diffuse in Kdm1a-ifKOs than in controls (Fig. 5B). Quantification of 2D plot images in CA1 neurons confirmed that ifKOs display smaller H3K27me3 puncta and a more dispersed signal, particularly noticeable around the chromocenters, compared to the control group (Supplementary Fig. 7A). However, there was no significant reduction in H3K27me3 levels (Supplementary Fig. 7B). We also compared the nuclear pattern in control and Kdm1a-ifKOs 6 and 18 months after TMX treatment (i.e., 8- and 20-month-old mice), and found that H3K27me3 disorganization progressed with age (Supplementary Fig. 7C). In fact, 20-month-old Kdm1a-ifKOs displayed a significant decrease in the intensity of H3K27me3 labeling (Fig. 5C and Supplementary Fig. 7B) in addition to more prominent dispersion of the signal surrounding the chromocenters.

We next investigated if the replacement of H3K27me3 by H3K27ac observed in ChIP-seq tracks of upregulated genes in Kdm1a-ifKOs (Fig. 2I) had a correlate in super-resolution images. The co-labeling with H3K27ac and H3K27me3 revealed a higher number of voxels in which the two signals coexist in the images from Kdm1a-ifKO than in those from control mice. (Fig. 5D, E). Like the aforementioned changes in the H3K27me3 distribution, the colocalization of the two marks also became more pronounced with age. Similar results were obtained when the overlap between H3K27me3 and H3K27ac was quantified using Mander's coefficient (Supplementary Fig. 7D). These data indicate that the epigenetic dysregulation unveiled in the multi-omic analysis is also reflected in alterations in subnuclear compartmentalization of epigenetic marks within neuronal nuclei.

### Kdm1a has liquid-liquid phase separation properties
The biophysics of liquid phases has recently emerged as critical for understanding nuclear compartmentalization[8]. For instance, recent studies indicate that altering liquid−liquid phase separation compromised the 3D genome organization without disturbing CTCF binding[49,50]. It has been also recently shown that many epigenetic regulators, including several KDM proteins, have LLPS-inducing

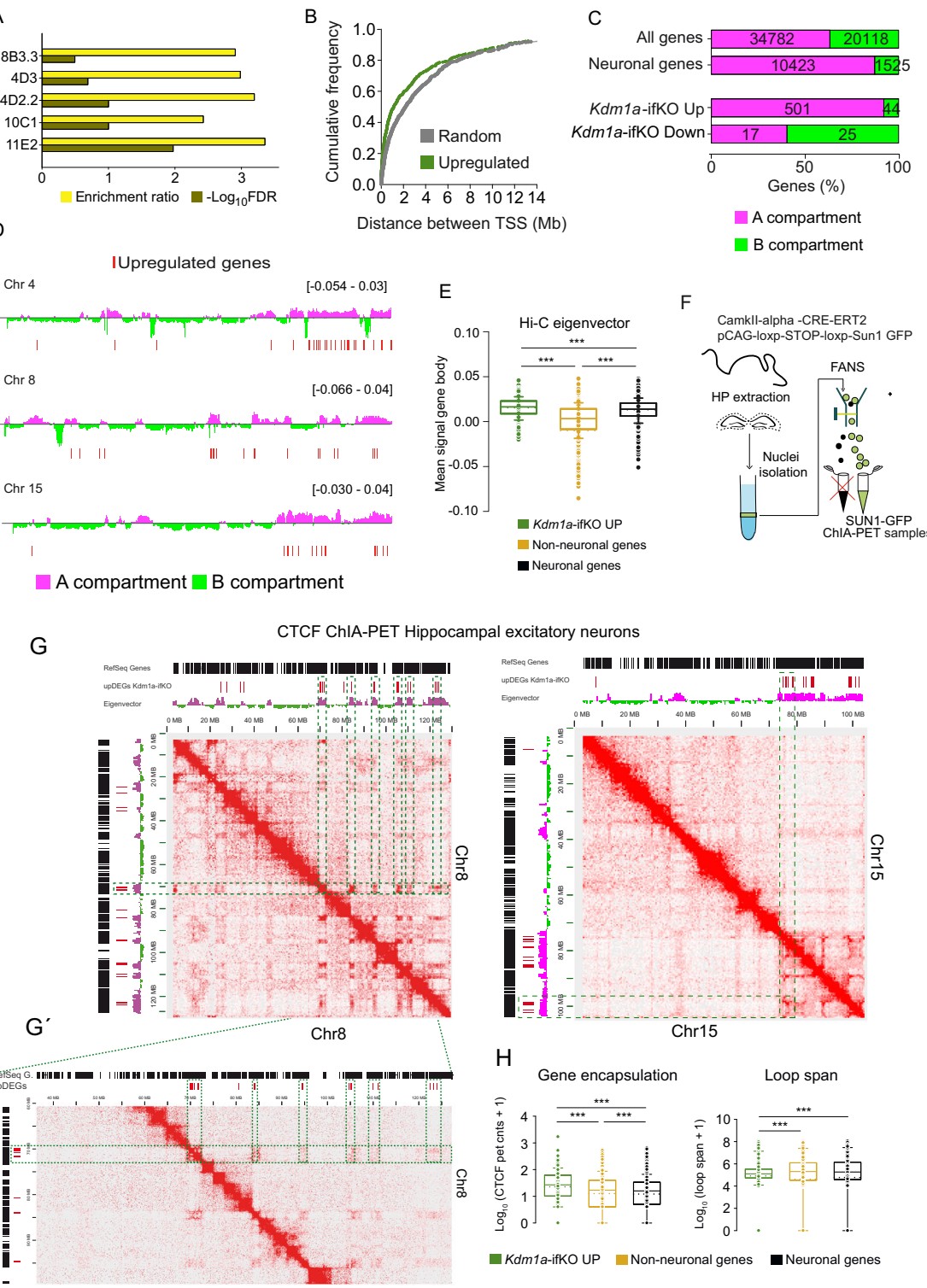

properties[51]. Therefore, we next explored if Kdm1a also has these properties. Several algorithms predicted that Kdm1a has an intrinsically disordered region (IDR) in its amino terminal (Fig. 6A, B, Supplementary Fig. 8A, B) a feature that is consistently found in phase separation proteins (PSPs). Furthermore, additional bioinformatical analyses indicated that Kdm1a has phase separation properties comparable to well-stablished PSPs, such as Mecp2, HP1alpha, MED1 and RNAPII (Supplementary Fig. 8C–E). To validate these predictions, we overexpressed human Kdm1a fused to GFP (Supplementary Fig. 8F) in HEK293 cells and found that it formed condensed droplets, which is a

feature of PSPs. These condensates showed fusion and fission events (Fig. 6C) and fluorescence recovery after photobleaching (FRAP) with a kinetic compatible with LLPS (Fig. 6D). Furthermore, the droplets rapidly disappeared after treating the cultures with 1,6-hexanediol, a compound with LLPS disruption properties (Fig. 6E, F)[52]. These results are consistent with recent experiments demonstrating that purified KDM1A protein formed phase-separated liquid compartments in vitro[53]. Therefore, although Kdm1a is not involved in CTCF binding or looping, it might alter phase distribution around the complex and, consequently, the topological interactions of upDEGs.

**Fig. 3 | Topological association between upDEGs. A** Chromosome location enrichment analysis of upDEGs (Kolmogorov–Smirnov test, p adj <0.05; $\log_2FC \geq 0.5$). Codes on the Y-axis correspond to the location of cytogenetic bands. **B** Cumulative plot reflects the distance between TSSs of upDEGs, and between TSSs of an equal number of random genes (Kolmogorov-Smirnov test, two-tailed, p adj <0.05; $\log_2FC > 0.5$; D = 0.14; p = 0.00004). **C** Percentage of genes located in compartment A or B for the genesets: all genes, neuronal and nonneuronal genes, and upregulated and downregulated genes in hippocampus of Kdm1a-ifKOs. To increase the scope of the analysis, we included the genes with $\log_2FC > 0.5$ (587 DEGs: 545 up and 42 down). Note that reads in the Hi-C dataset were binned at 25 kb resolution. **D** Genomic snapshots of upDEGs (red lines) clustering in chromosomes 4, 8, and 15. The location into the A/B compartment is shown. Selection criteria for non-neuronal and neuronal genes were TPM < 5 (14902 genes) and TPM > 5 (11372 genes), respectively, according to data from[26] **E** Hi-C average interaction frequency[26] within the gene body of upDEGs, genes expressed in neurons and nonneuronal genes not affected by Kdm1a loss (Mann-Whitney U Statistic, two-tailed, pval ***<0.0001). Box plots indicate median value, interquartile range, minimum and maximum value, and individual data points. **F** Scheme of isolation of Sun1-GFP tagged nuclei. **G–G'** 2D visualization of CTCF-dependent chromatin interaction at excitatory hippocampal neurons from WT mice. Intrachromosomal interactions at Chr5 (0Mb-104Mb) and Chr8 (0Mb-120Mb) are shown. Black bars: referenced genes at chr8 or chr15. Green dashed line boxes: intrachromosomal loops between upDEGs. **H** Boxplots present the quantification of CTCF interaction in the sets of upDEGs, nonneuronal genes and neuronal genes. The left panel shows the summation of ChIA-PET counts at CTCF loops encompassing the gene body. The right panel shows the loop span measured as the average distance between the CTCF peaks that encapsulate the upDEGs (Mann–Whitney U Statistic, two-tailed, pval ***<0.0001). Box plots as indicated in panel E. Source data are provided as a Source data file.

Next, to investigate if the PSP property of Kdm1a is necessary for the derepression of Kdm1a's target genes, we turned to primary neuronal cultures (PNCs) produced from *Kdm1a^{f/f}* embryos (Fig. 6G). First, we verified through RT-qPCR assays that ablation of Kdm1a in hippocampal PNCs infected with a Cre-recombinase-expressing lentivirus (Supplementary Fig. 8G, H) displayed the activation of upDEGs identified in our RNA-seq screen in adult Kdm1a-ifKOs (Fig. 6H). Then, we tested whether the expression of a full-length human isoform of Kdm1a (lacking the alternative exon 2a and the neuron-specific mini-exon 8a) prevented the activation of non-neuronal genes. RT-qPCR analysis demonstrated a full rescue of the transcriptional phenotype (Fig. 6I, J). Notably, the expression of a truncated version of the same protein lacking the amino-terminal IDR (Supplementary Fig. 8I–K) did not avoid gene derepression (Fig. 6I, J). In contrast, several previously described mutations that affect the demethylase activity of KDM1A had lower or no impact in the activation of non-neuronal genes compared to the IDR deletion mutant (Supplementary Fig. 8L). These results highlight the critical role of the IDR in Kdm1a's nuclear compartmentalization and repressive functions (Fig. 6K).

### Kdm1a-related gene derepression naturally occurs in aging mice and humans

To investigate if the progression of chromatin alterations with age observed in super-resolution images correlated with stronger changes in transcription, we analyzed the expression of upDEGs in young adult and elderly mice. The hippocampus of 20-month-old Kdm1a-ifKOs showed stronger activation of upDEGs (Fig. 7A), consistently with the exacerbation of epigenetic dysregulation with age observed in super-resolution images. Interestingly, a weak but significant upregulation of upDEGs was also observed in the hippocampus 20-month-old control mice, suggesting that the dysregulation of these genes may be a feature of brain aging. This result is consistent with a recent study showing that the euchromatinization of H3K27me3-repressed genes is a feature of aging tissues[54].

We next investigated if the age-dependent derepression of PRC2-repressed genes observed in elderly mice was also observed in humans. We found that the homolog genes of Kdm1a-ifKO upDEGs were enriched in H3K27me3 both in adult and fetal human brain samples (Fig. 7B). Furthermore, the analysis of three large datasets exploring age-related transcriptional changes in the human brain (GSE33000, GSE15222 and GSE48350) revealed a positive correlation between transcript levels and age in the group of upDEG ($\log_2FC > 1$) homologs (R = 0.28, p = 3.4e-07), indicating that these nonneuronal genes are also derepressed during aging in humans (Fig. 7C and Supplementary Data 4). Moreover, several of the genes exhibiting stronger age-dependency are marked with H3K27me3 (Fig. 7C, D), mapped together within regions with conserved synteny between mouse and human genomes (e.g., *NASG, OTOP2, TMC8, TK1* and *SLC16A3* gene cluster) (Fig. 7E, F) and interact with each other through CTCF loops

(Supplementary Fig 9A, B). These observations suggest that the epigenetic changes observed in Kdm1a-ifKOs may naturally emerge during aging both in humans and in mice.

## Discussion

Kdm1a plays a critical role during the development of the nervous system and is still the most expressed histone demethylase in mature neurons. However, its function in the maintenance of neuronal identity and epigenetic profiles in the adult brain remains unknown. Most functional studies of Kdm1a have been carried out in tumor cells[27] with few investigations focusing on its role in non-mitotic cells such as neurons. Previous studies investigated the role of the neuronal Kdm1a isoform (nKdm1a) during neuronal differentiation[19] and described the consequences of Kdm1a deficiency using conventional knockout mouse strains[55] or early gene ablation during development[13]. Therefore, these studies could not dissect the developmental and adult functions of Kdm1a. To specifically investigate the role of Kdm1a in mature neurons of the adult mouse brain, eliminating the confounding effect of Kdm1a elimination at earlier stages, we specifically eliminated Kdm1a in 2-month-old mice or older using the *Camk2-Cre-ERT2* tamoxifen-inducible recombination system. Our investigation on inducible, neuron-specific knockouts demonstrated that Kdm1a is not required for the survival of forebrain principal neurons. We did not detect any sign of neuronal death even one year after precise gene ablation in mature principal neurons, nor the upregulation of genes related with apoptosis, death or inflammation in their transcriptome and epigenome analyses. These results sharply contrast with the conclusions in a previous study indicating that Kdm1a cell-autonomously protects neurons against degeneration[21]. That article investigates a strain of mice (*Kdm1a^{CAGG}*-KO) in which the CAG promoter drove Kdm1a ablation in most tissues and cell types leading to paralysis, widespread hippocampal and cortical neurodegeneration and premature death[21]. Considering the key role of Kdm1a in the differentiation of many cell types, the severe phenotype observed in *Kdm1a^{CAGG}*-KOs can be caused by the loss of self-renewing cells, such as astroglia and endothelial cells, that support and provide nutrients to neurons.

Our investigation on inducible and neuron-specific KOs indicates that Kdm1a is needed throughout the whole life of the animal to maintain the repressed state of a set of genes that show low or no expression in wild type mature neurons. The genes derepressed in neurons after Kdm1a loss are decorated by the facultative repressive mark H3K27me3 and bound by PRC2. The repressor role of Kdm1a described in this study is consistent with previous research in other cell types and model organisms. For instance, studies in mouse gastrula have shown that endogenous MuERV-L/MERVL retroviruses as well as cellular genes flanked by MERVL sequences, whose expression is normally restricted to the zygotic genome activation (ZGA) period, become up-regulated in the absence of Kdm1a[56]. Furthermore, the Kdm1a-containing CoREST complex exerts an important function in

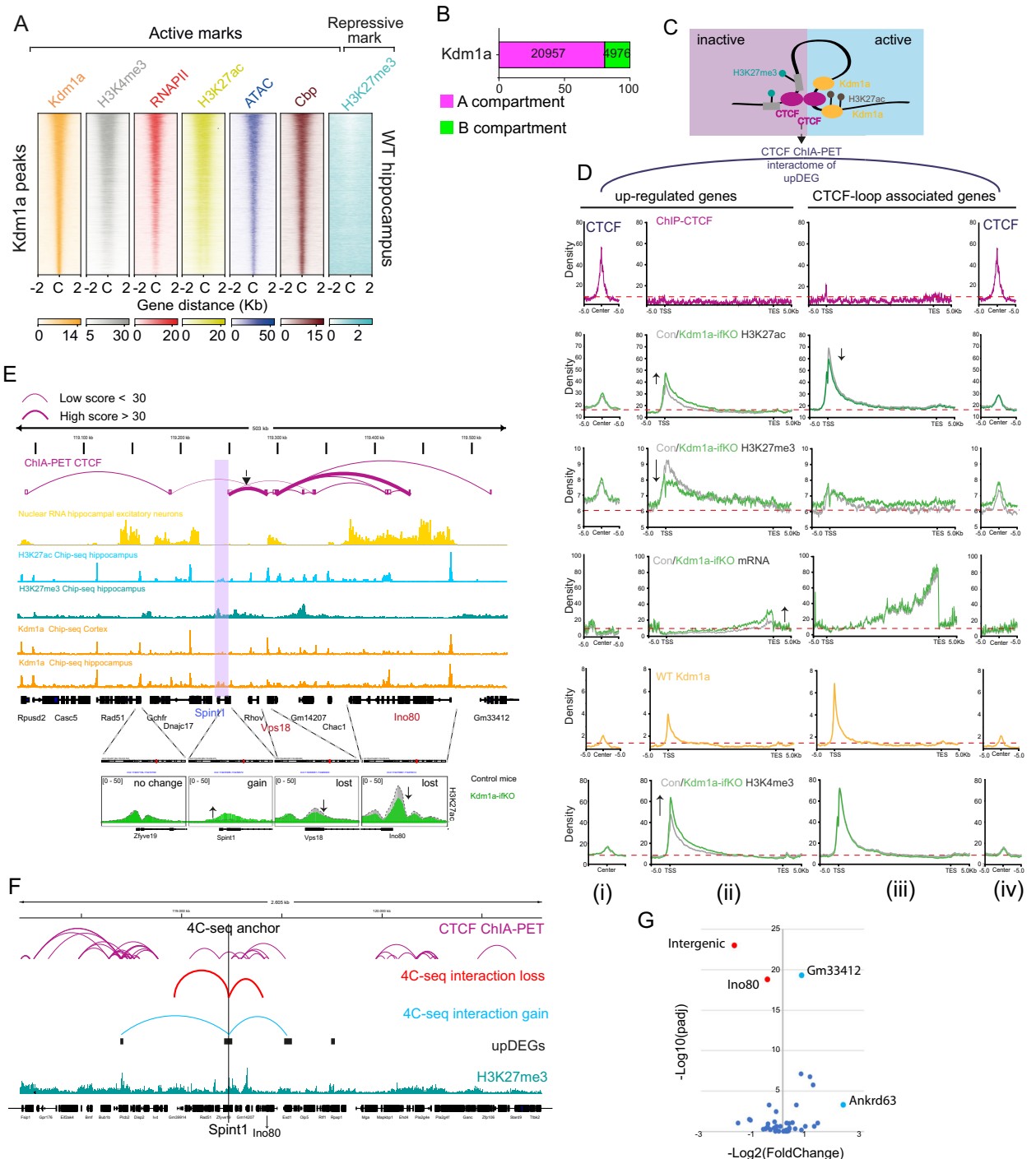

**Fig. 4 | Topological association between Kdm1a and its target genes.**
**A** Heatmaps of active and repressive marks in the chromatin of WT mice ( ± 2 kb of Kdm1a peaks). **B** Distribution of Kdm1a peaks between chromatin compartments. **C** Schematic illustration of CTCF connection in upDEGs. **D** Density plots showing ChIP-seq signal for H3K27ac, H3K27me3 and H3K4me3 in CTCF loops associated with upDEGs in hippocampal chromatin of control and Kdm1a-ifKO mice. Four graphs are presented for each hPTM corresponding to (i) ±5 kb window centered on the CTCF peaks directly linked to the upDEGs; (ii) upDEGs metagene; (iii) metagene for the CTCF-loop associated genes; and (iv) ±5 kb window centered on the CTCF peaks directly linked to the upDEG-associated genes. Transcript levels in control and ifKO mice, and CTCF and Kdm1a density plots in WT hippocampal chromatin are also shown. **E** Genomic profiles and CTCF ChIA-PET interactions at the *Spint1* locus (purple box). CTCF loops bring the H3K27me3-enriched *Spint1* gene closer to active genes *Vps18* and *Ino80*. The lower panels show H3K27ac levels in the chromatin of control and Kdm1a-ifKOs at *Spint*1 and neighbour genes linked

through first and second-order CTCF connections. *Vps18* and *Ino80* show lower levels of H3K27ac in Kdm1a-ifKOs, in parallel to the increase for this hPTM in *Spint1*. The dashed line represents H3K27ac density in control mice. Transcript levels are shown in the yellow track. **F** 4C-seq interaction profiles using the *Spint1* promoter as viewpoint (anchor). Interactome profiles were generated using the hippocampus of Kdm1a-ifKO (*n* = 3) and control mice (*n* = 3). Significant reduction in the interaction of *Spint1* was observed with the gene body of *Ino80* and with an intergenic region (in red), while a higher interaction frequency was observed with *Gm33412* and *Ankrd63* (in blue), which are also upregulated in Kdm1a-ifKO neurons. The position of upDEGs is indicated with black boxes; the tracks for H3K27me3 ChIP-seq and CTCF-ChIA-PET interactions are shown. **G** Volcano plot showing the most significant changes on interaction frequencies in the 4C-seq generated with *Spint1* as viewpoint (likelihood ratio test, pval <0.1). Source data are provided as a Source data file.

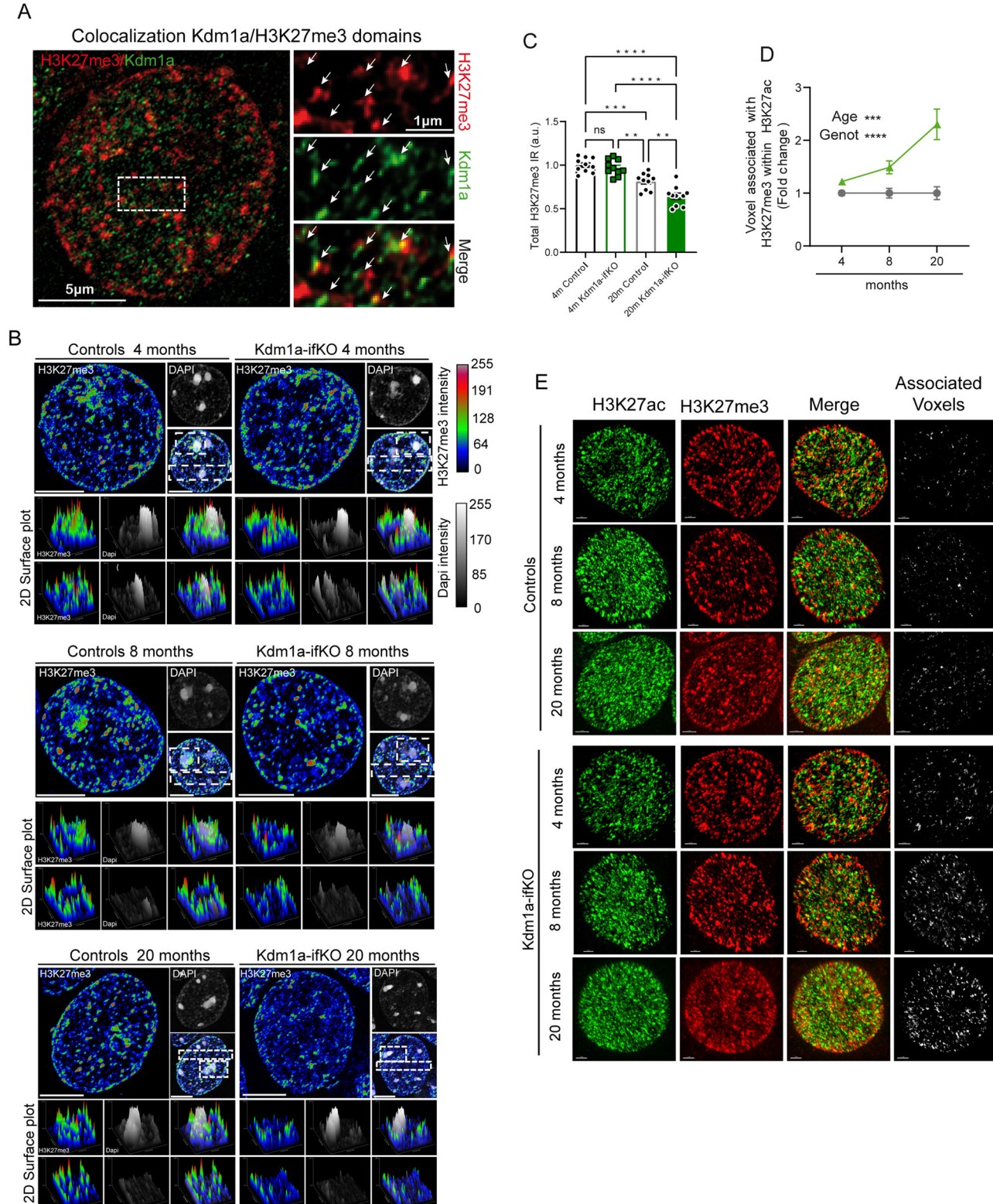

repressing neuronal genes in nonneuronal cells[57]. These results contrast with the postulated role of the splicing isoform known as neuroLSD1 as a transcriptional activator[13,55], suggesting that this variant exerts its transactivation function during neuronal specification and maturation[19], although its absence may still have important consequences at later stages[58].

Although not described before in mammals, the role of Kdm1a as a regulator of boundaries between silenced and active chromatin

domains has been previously postulated in flies and fungi[14,15,59,60]. Our experiments shed light on how Kdm1a exerts this regulatory function. Contrary to our expectations, the binding of Kdm1a (as determined by ChIP-seq data) did not serve as a reliable predictor of transcriptional dysregulation in ifKOs. In fact, the majority of upregulated genes did not directly bind Kdm1a, and most Kdm1a-bound genes remained unaffected by the ablation of Kdm1a, suggesting that other KDMs collaborate with Kdm1a in active genes to maintain proper levels of lysine

**Fig. 5 | Altered chromatin compartmentalization in Kdm1a-ifKOs progresses with age. A** Super-resolution images show the colocalization of Kdm1a and H3K27me3-immunoreactivity in the nucleus of CA1 pyramidal neurons in WT mice. **B** Representative super-resolution images show the disorganization of H3K27me3 condensed aggregates in the nucleus of Kdm1a-ifKO neurons from 4- (top panels), 8- (middle panels) and 20-month-old (bottom panels) mice compared to control littermates. The distribution of H3K27me3 puncta at selected regions (white boxes) is shown as a 2D surface plot in the panels below. Fluorescence intensity of H3K27me3 and DAPI labeling is represented with a color gradient. The red signal indicates the highest detected H3K27me3 signal intensity and the blue the lowest. Scale bar: 5 μm. **C** Measurement of H3K27me3-immunoreactive signals in the nucleus of CA1 hippocampal neurons ($n = 10$) of 4-, 8- and 20-month-old Kdm1a-ifKO and control littermates (4 m, $n = 3$; 20 m, $n = 4$ per genotype). Tukey's multiple comparison test, two-sided, \*\*\*\*$p$-val < 0.0001; \*\*\*$p$-val < 0.001; \*\*$p$-val < 0.01. **D** Quantification of association between H3K27me3-positive voxels and H3K27ac signal in Kdm1a-ifKOs and control mice at different ages (4 m, $n = 15$ cells; 8 m, $n = 10$ cells; 20 m, $n = 11$ cells). The quantification of voxels with overlapping signals revealed that the proportion of H3K27me3-positive voxels with H3K27ac signal increased with age in Kdm1a-ifKOs compared to control mice (4 m, $n = 3$ mice; 8 m, $n = 2$ mice; 20 m, $n = 4$ mice per genotype; two-way ANOVA: \*\*\*\*$p$-val < 0.0001, \*\*\*$p$-val < 0.001). **E** Representative super-resolution images show the increased coincidence of H3K27me3 and H3K27ac signals in the nucleus of Kdm1a-ifKO neurons from neurons from 4-, 8- and 20-month-old mice compared to control littermates. Scale bar: 2 μm. Data are presented as mean values ± SEM in (**C**) and (**D**). Source data are provided as a Source data file.

methylation. However, we observed a significant increase in the number of upregulated genes that interacted with Kdm1a when CTCF-mediated interactions were considered, suggesting that this protein may contribute to establishing or maintaining three-dimensional chromatin boundaries. The widespread changes in H3K4me3 and their robust correlation with transcriptional changes strongly suggest the participation of Kdm1a's catalytic activity in this process. The disruption of H3K4 methylation balance resulting from the loss of H3K4me2/H3K4me1 demethylation can lead to the increase in H3K4me3, and this, in turn, causes a reduction in H3K27me3 and an elevation in H3K27ac, ultimately leading to the derepression of genes. Similar observations have been made in other cell types upon Kdm1a pharmacological inhibition or gene elimination[56]. For instance, pharmacological and genetic experiments in *Drosophila* have revealed a direct relationship between H3K27ac levels and the activity of Kdm1a homolog (encoded by the *Su(var)3-3* gene)[61]. Kdm1a's scaffolding and LLPS functions are also likely crucial for its repressive role. Supporting this hypothesis, our experiments show that the loss of Kdm1a destabilized H3K27me3 micro-domains, resulting in compromised boundaries between active and repressed genes. In addition to Kdm1a, it is plausible that other epigenetic regulators are also involved in topological associations with their target genes, providing an explanation for their influence on distantly located targets beyond what is evident in linear genome profiles.

Seminal studies in the PSPs field demonstrated that both H3K27ac-decorated super-enhancers and H3K9me3-decorated constitutive heterochromatin can form phase condensates[62,63]. More recently, it was proposed that PRCs can also form liquid phase separation[29] and that H3K27me3-rich regions interact preferentially with each other and their interaction may give rise to phase condensates[30]. Nichols and Corces have recently shown using high-resolution Hi-C data and computer simulations that chromatin compartments arise because of proteins interactions that correlate with the presence of H3K27ac, H3K27me3, and H3K9me3[64]. Adding to this picture, our results indicate that Kdm1a contributes to isolating adjacent H3K27ac and H3K27me3 microdomains. The structural properties of Kdm1a may place this protein in the interface of H3K27ac and H3K27me3 where it prevents the spreading of the active mark into silenced loci (this study[15]). In turn, possibly in other cell types or genomic contexts, Kdm1a might also prevent the spreading of repressive marks into active loci[14,60]. This ambivalent behavior could be explained by the promiscuous substrate specificity of Kdm1a depending on its interacting partners[12].

Recent studies indicate that blocking transcription from cryptic promoters is critical for establishing precise cell type-specific gene activation[65]. Therefore, not surprisingly, the derepression of non-neuronal genes is associated with more severe phenotypes when Kdm1a is removed early in life[16,17]. Like for other epigenetic regulators[66–68], the importance of Kdm1a repression decreases at later stages when repressed genes acquire a permanent silent status by the action of alternative epigenetic mechanisms, such as DNA methylation, that lead to the heterochromatinization of the loci[69]. The presence of

ectopically or abnormally produced transcripts observed in the hippocampus of Kdm1a-ifKOs could potentially disrupt neuronal function in unforeseen manners, thereby contributing to Kdm1a-associated intellectual disability[70].

The role of Kdm1a in preserving topological boundaries not only has the potential to advance treatments and corrective strategies for CPRF syndrome but also holds promise for improving therapy for other Kdm1a-related disorders that emerge during postnatal stages. This includes various cancers with poor prognosis, where KDM1A is highly expressed and considered a potential target for cancer therapy[27,71]. Additionally, in the analysis of post-mortem brain samples from patients with Alzheimer's disease (AD) and frontotemporal dementia (FTD), mislocalized KDM1A immunoreactivity was detected in cytoplasmic tangle-like aggregates and neurites[21]. Recent studies have also suggested that inhibiting Kdm1a might contribute to neurodegeneration in a mouse model of tauopathy[72]. Although we did not detect any sign of neuronal death in Kdm1a-ifKOs even one year after Kdm1a ablation, our experiments comparing elderly and young adult mice suggest that Kdm1a may play a role during neuronal aging, preserving chromatin domain boundaries and preventing the epigenetic erosion that has been postulated as a major landmark of the aging process[73]. Consistent with our findings in brain tissue, a recent study comparing epigenetic profiles in the liver of young and old mice reported the euchromatinization of H3K27me3-microdomains contained in the A compartment[54], pointing out that this process could also occur in other tissues during aging[74]. We, however, did not detect a significant increase of cytoplasmic Kdm1a levels or reduction of nuclear Kdm1a in principal neurons of 20-month-old mice. Future investigations should further explore the role of Kdm1a and other epigenetic regulators in LLPS in neuronal nuclei and its relevance in age-related epigenetic dysregulation, cognitive decline and brain aging.

## Methods
### Mouse strains and treatments
*Kdm1a*^f/f [75], CAG-Sun1/sfGFP[76] and *Camk2a*-creER^T2[77] mice are available at public repositories (Jackson Lab stock #23969, MMRRC stock 066789-UCD; Jackson Lab stock #021039, RRID:IMSR_JAX:021039 and EMMA EM:02125, respectively). All mice were maintained on a C57BL/6 J genetic background. Mice were maintained and bred under standard conditions, consistent with Spanish and European regulations. All the protocols for animal experimentation were approved by the Animal Welfare Committee at the Instituto de Neurociencias, the CSIC Ethical Committee and the Dirección General de Agricultura, Ganadería y Pesca of Generalitat Valenciana. In detail, mice were maintained under specific pathogen-free (SPF) conditions within the Animal House at the Instituto de Neurociencias (CSIC-UMH), in a 12 h light/12 h dark cycle (7:00 a.m. to 7:00 p.m.) at 20–24 °C and controlled humidity (40-60%) with free access to food and water. *Kdm1a*^f/f mice bear loxP sites flanking exons 5 and 6; their recombination results in the appearance of a premature stop codon in the mRNA. *Camk2a*-creER^T2 x *Kdm1a*^f/f

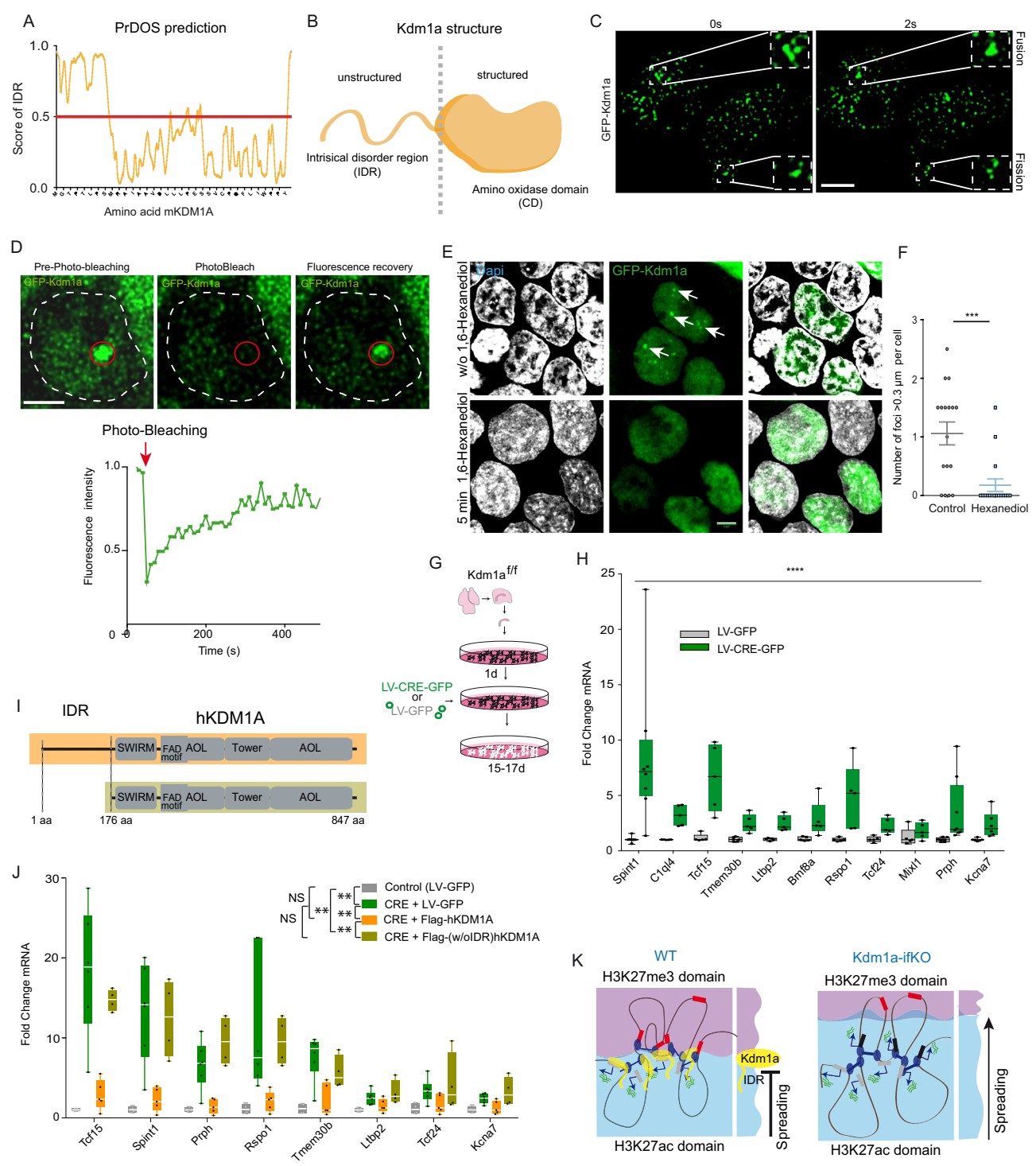

mice selectively eliminate *Kdm1a* in principal neurons of the forebrain after TMX administration in 8-week-old mice. *Camk2a*-creER[T2] x CAG-Sun1/sfGFP were used to isolate principal neurons nuclei of the forebrain in 12-week-old mice after one month of TMX administration. The recombination of floxed alleles was induced by 5 intragastrical administrations of TMX (Sigma Aldrich, 20 mg/mL dissolved in corn oil) on alternate days. In all our experiments with ifKOs, we used CaMKIIα-creERT2- littermates treated with tamoxifen as controls. For phenotypic analyses, young adult (3-month-old, 1-month after TMX), adult (9-month-old, 7 months after TMX) and aged (18-month-old, 16 months after TMX) mice were used as indicated. For the study of the response to activity of *Kdm1a*-ifKO neurons, 18-month-old mice were

intraperitoneally administered 25 mg/Kg of kainic acid (KA) and sacrificed 1 h later.

**Immunohistochemistry and cell death assays**

Mice were anesthetized with a mix of ketamine/xylacine and perfused with 4% paraformaldehyde in PBS. After an overnight postfixation, coronal vibratome sections (50 μm) were obtained, washed in PBS and PBS − 0.1% Triton X-100 (PBT) and incubated for 1 h at room temperature with 3-5% heat inactivated newborn calf serum in PBT. The primary antibodies used were α-Kdm1a (ab17721, 1:200 or 1:1000), α-NeuN (Chemicon MAB377, 1:500), α-GFAP (G3893, Sigma, 1:500), α-Cleaved Caspase-3 (9661, Cell Signaling, 1:500), α-Fos (226004,

**Fig. 6 | Kdm1a condensates exhibit LLPS properties. A** The PrDOs algorithm identified intrinsically disordered regions at the N-terminus of mouse Kdm1a. **B** Kdm1a scheme illustrating the separation of the IDR and catalytic domain. **C** Time-lapse images of the nuclei of HEK293 cells expressing GFP-Kdm1a. Fusion and fission events of Kdm1a puncta were captured in a 2 second interval. Scale bar: 5 μM. **D** GFP-Kdm1a expression in HEK293 cells produces green-fluorescent puncta with rapid fluorescence recovery after photobleaching (FRAP) (*n* = 4 independent experiments) Scale bar: 2.5 μM. **E** GFP-Kdm1a droplets disappeared few minutes after treatment with 1,6-hexanediol. Scale bars: 5 μM. **F** Quantification of GFP-Kdm1a foci > 0.3 μm per cell before and after 1,6-hexanediol treatment. (*n* = 17 cells in 3 independent experiments; Mann-Whitney test, two-tailed: ***pval = 0.0004). Scatter plot represents mean ± SEMs. **G** Scheme of experiments in hippocampal PNCs. E17 *Kdm1a ᶠ/ᶠ* embryos infected with a lentiviral vector co-expressing CRE recombinase and GFP (LV-CRE-GFP) or a control lentiviral vector that only expresses GFP (LV-GFP). The same elements (created by B.dB.) in a different order were used in[80] (Lipinski et al., KAT3-dependent acetylation of cell type-specific genes maintains neuronal identity in the adult mouse brain. *Nat Comm* 2020 May 22;11(1):2588. https://doi.org/10.1038/s41467-020-16246-0). **H** RT-qPCR analysis

demonstrates that de-repression of upDEGs also occurs in Kdm1a-deficient PNCs. (control, *n* = 3; CRE, *n* = 3; two-way ANOVA: ****p-val < 0.0001). Boxplots indicate median value, interquartile range, minimum and maximum value, and individual data points. **I** Schematic representation of the human Kdm1a variants used in rescue experiments. **J** RT-qPCR analysis demonstrates that hKDM1A protein restores non-neuronal genes repression in Kdm1a-deficient PNCs. However, a truncated KDM1A protein lacking the N-terminal IDR is unable to restore the transcriptional phenotype found in Kdm1a-deficient PNCs. (control, *n* = 4; CRE, *n* = 6; Flag-hKDM1A, *n* = 5 and Flag-(w/oIDR)hKDM1A, *n* = 4; two-way ANOVA: **p-val2 < 0.001; NS pval >0.01). Boxplots as indicated in panel H. **K** Schematic model of Kdm1a function at the boundary of H3K27me3 and H3K27ac domains. The loss of Kdm1a causes changes in H3K4 methylation levels, alters the balance between H3K27me3/H3K27ac and disrupts chromatin compartmentalization leading to the de-repression of PCR2-repressed nonneuronal genes that interact with active loci. Kdm1a's IDR is essential to maintain non-neuronal repression, suggesting that liquid-liquid phase separation is involved in the transcriptional and epigenetic alterations described in Kdm1a-ifKOs. Source data are provided as a Source data file.

Synaptic Systems, 1:500), α-Parvalbumin (P3088, Sigma-Aldrich, 1:500), α-Pecam 1 (550274, BD Pharmingen™, 1:500), α-H3K27me3 (07-449, Millipore, 1:500), α-H3K27me3 (ab6002, Abcam, 1:250) and α-H3K27ac (ab4729, Abcam, 1:500). Nuclei were counterstained with a 1 nM DAPI solution (Invitrogen). For the TUNEL assay, we used the in situ Cell Death Detection Kit (11684795910, Roche) following the manufacturer's instructions. DAB staining was performed according to the instructions of the product (11718096001, Roche). Images were taken with an Inverted Confocal Microscope Olympus FV1200 and Zeiss LSM 880.

## Confocal and super-resolution microscopy

For super-resolution imaging, nuclei were imaged using a Confocal Microscope Zeiss LSM 880-Airyscan Elyra PS.1, equipped with 63x PlanApo oil-immersion objective lens (NA 1.4) and processed with ZEN 2.3 software (Zeiss, RRID: SCR_013672). Optimal image size and pixel/voxel size (distance between planes) were 1912×1912 pixels and 0.18 μm, respectively. All super-resolution images were then deconvolved with Huygens Professional version 21.10 (Scientific Volume Imaging, The Netherlands, http://svi.nl, RRID: SCR_014237). For surface plot analysis, images were processed as Maximum Intensity Projection along the z-axis with Fiji software (ImageJ 1.53f51, RRID: SCR_002285) and analyzed using 'Surface Plotter' tool at selected areas. To quantify/analyze H3K27me3 distribution, super-resolution and deconvolved nuclei were maximum projected along the z-axis before segmentation using Mean's method from Fiji, which resulted in a binary mask that shows H3K27me3 presence (in white the pixels belonging to any H3K27me3 signal) or absence (in black the pixels belonging to the background). Sholl analysis v4.0.0 plugin (of Fiji software) was performed on the segmented nuclei creating concentric circles from nucleus center to periphery using a starting radius of 1 μm up to 8 μm and 0.049 μm radius step size; the number of intersections with the binary signal of H3K27me3 were plotted, fitted using 7th degree polynomial function. To analyze the spatial distribution of H3K27me3 and *Kdm1a* in the nucleus, selected planes from the stack were obtained with ZEN 2.3 software. The "Measure distances" task, from Imaris 8.2.0 software (Bitplane, RRID:SCR_007370), was used to automatically measure the distance between two proteins/signals from centroid to centroid in the 3D image, on Surpass objects (Spots in that case). The centroid of the object is calculated by taking the average position of the voxels making up the object. The histone marks colocalization/overlapping analysis was made using automatic thresholding within the 'Coloc' tool in Imaris 8.2.0 software on the 3D image. Confocal images were obtained using a vertical Confocal Microscope Leica SPEII with oil-immersion 63x objective lens (NA 1.4), with a 1024×1024 collection

box. Confocal pinhole was set to 1 AU for each channel and pixel/voxel size was 0.8 μm. Images were processed as Maximum Intensity Projection with Fiji software and analyzed using 'Plot Profile' tool at selected lines that crossed at least two chromocenters. Total immunoreactivity was measured as the sum of intensities of all pixels inside each nucleus. Crosstalk between fluorophores was eliminated by adjustment of the spectral ranges of the detectors and sequential scanning of the images.

## Fluorescence-activated nuclei sorting (FANS)

Mice were sacrificed by cervical dislocation. After hippocampal dissection, cell dissociation and lysis was performed by mechanical homogenization in a 2 mL dounce homogenizer (Sigma-Aldrich) filled with 1 mL Nuclei Extraction Buffer (250 mM Sucrose, 25 mM KCl, 5 mM MgCl₂, 20 mM Hepes-KOH, 65 mM β-glycerophosphate, 0,5% IGEPAL CA-630, 0,2 mM Spermine, 0,5 mM Spermidine, Proteinase inhibitors (complete EDTA-free, Roche)). Using 35 μm mesh-capped tubes, nuclei were filtered to get rid of debris. Incubation with 0.01 mM DAPI. For the clean isolation of nuclei, samples were diluted in a 22% Optiprep medium (D1556, Sigma-Aldrich) solution, and a density gradient was prepared in 15 mL centrifuge tubes as follows: 2 mL 44% Optiprep, 1 mL 22% Optiprep, 22% sample containing the nuclei, and additional 22% Optiprep to fill the tube. The gradient was centrifuged for 20 minutes at 10000 x g and the phase containing the nuclei was collected. Nuclei were then sorted in a flow-cytometer FACS Aria III (BD Bioscience) and GFP positive nuclei were selected according to the parameters defined in ref. [26] (see step by step analysis and channel filtering details in Supplementary Fig. 4H).

## 4C-seq

Circular chromosome conformation capture, coupled to high-throughput sequencing (4C-seq) was performed as previously described[78] with minimal changes. Approximately 5 million hippocampal nuclei were fixed at 1 % with PFA. The restriction enzyme cutting order was Dpn II as the primary enzyme and Csp6I as the secondary enzyme. Ligated chromatin was phenol-chloroform extracted and ethanol precipitated. Reactions were then further purified with Ampure XP beads with a 0.8x ratio of bead solution to library following the manufacturer's instructions. Samples were then quantified with Qubit and 4C-seq libraries were sequenced using an Illumina high-throughput sequencing equipment. For 4C-seq analysis, all samples were trimmed to 75 bp. The wig and peaks were calculated using *pipe4C* and *peakC* as mentioned in[78]. We removed sample WT1 due to a batch effect. Call for differential contact frequencies and PCA were performed using Deseq2 after batch effect removal.

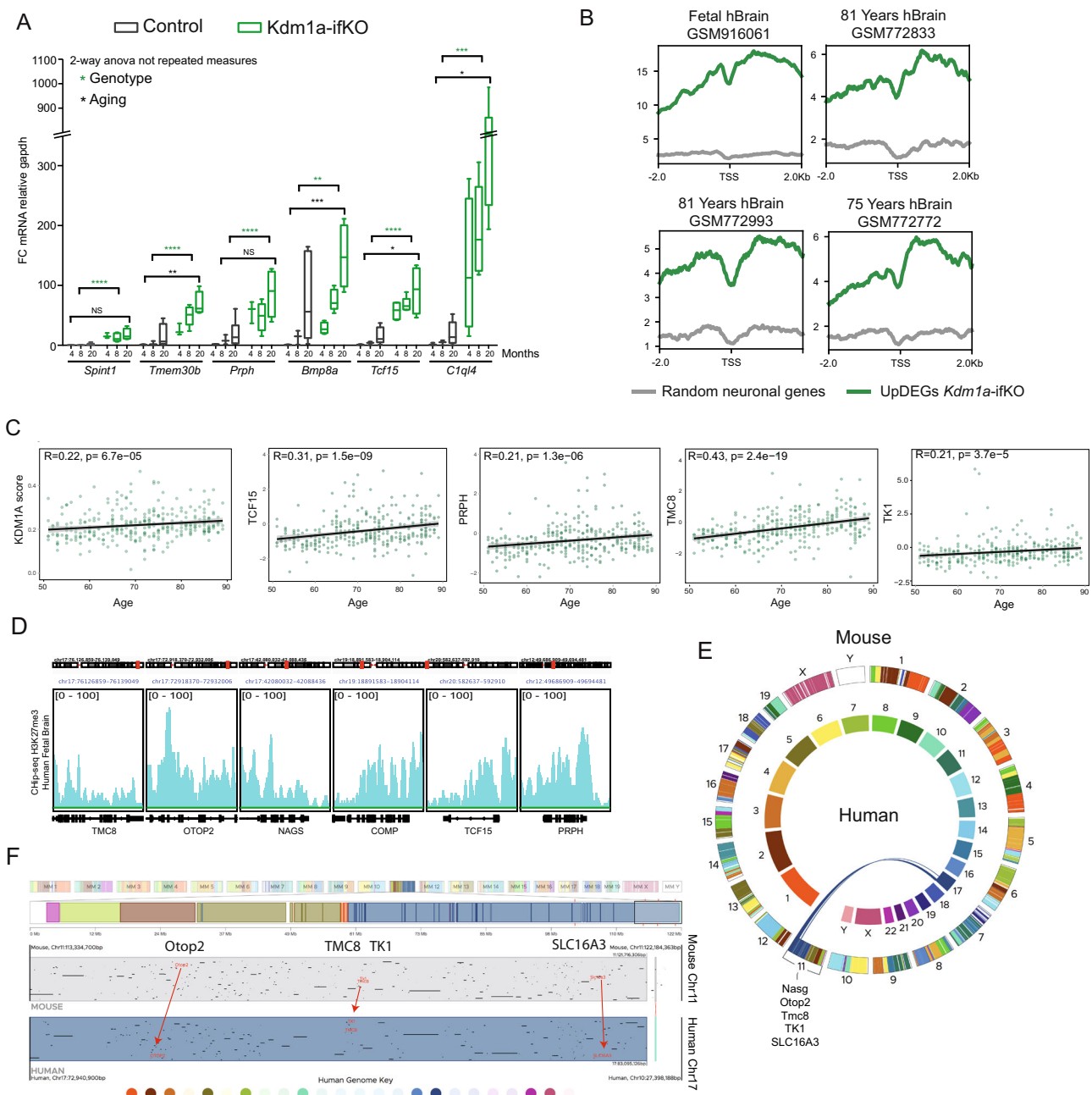

**Fig. 7 | Derepression of upDEGs both in mice and humans correlates with aging.**
**A** Time course analysis of selected upregulated genes, *Prph, Tmem30b, Spint1, Bmp8a, Tcf15* and *C1ql4* of Kdm1a-ifKO neurons compared with control neurons using RT-PCR (4 m, $n = 3$; 8 m, $n = 3$; 20 m, $n = 6$ in controls; and 4 m, $n = 4$; 8 m, $n = 5$; 20 m, $n = 4$ in Kdm1a-ifKOs). Boxplots indicate median value, interquartile range, minimum and maximum value, and individual data points. Two-way ANOVA: ****$p$-val < 0.0001; **$p$-val < 0.001; *$p$-val < 0.01; NS pval >0.01. **B** Density plot of H3K27me3 in adult and fetal human brain samples (datasets used and ages are indicated in the figure). H3K27me3 was plotted at the TSS of the human homolog genes of upDEGs in Kdm1a-ifKOs ($\log_2$FC > 1). **C** Gene expression correlation between age and transcript level of Kdm1a-ifKO upDEGs ($\log_2$FC > 1.5) ($n = 320$).

Kdm1a score was calculated using the Kdm1a-ifKO genes represented in the human dataset (100 out of 165 genes). Spearman Correlation, pval <0.05. Several examples of genes with a high and intermediate degree of correlation are shown. **D** Genome snapshot of H3K27me3 enrichment in fetal brain samples for several genes upregulated during human and mouse aging: *TMC8, OTOP2, NAGS, COMP, TCF15,* and *PRPH*. **E**, **F** Analysis of regions of conserved synteny using the web-based application The JAX Synteny Browser[104]. Chromosome position analysis of *Nags, Otop2, Tmc8, Tk1,* and *Slc16a3* in the human and mouse genomes (**E**). Representation of human-mouse synteny across the entire domain spanning *Otop2, Tmc8, TK1,* and *Slc16a3* (chr17 and chr11, respectively in human and mouse) (**F**). Source data are provided as a Source data file.

## Western blot
Hippocampal tissues were harvested and homogenized. The protein samples were mixed with 5 × sodium dodecyl sulfate-polyacrylamide gel electrophoresis sample buffer (2 μg/μL), boiled for 5 minutes and sonicated. The samples were separated by 12% sodium dodecyl sulfate polyacrylamide gel electrophoresis, electrotransferred to

nitrocellulose filters, blocked with 5% defatted milk at room temperature for 1 hour, and incubated the following primary antibodies: α-SUZ12 antibody (ab175187 Abcam, 1:500) and α-β-actin monoclonal antibody (clone AC-74, Merck, 1:500). The membranes were washed three times with PBS/Tween-20, incubated with secondary antibody goat anti-rabbit IgG-horseradish peroxidase (Sigma) for 60 minutes,

washed three times with PBS/Tween-20, and visualized by enhanced chemiluminescence using an Amersham™ Imager 680 apparatus. β-actin signal was used for normalization.

## RT-qPCR analyses

For RT-qPCR, total RNA was extracted from hippocampal tissue with TRI reagent (Sigma-Aldrich) as recommended by the manufacturer. Subsequently, genomic DNA was eliminated by treatment with DNAse I (Qiagen) for 30 min at 25 °C. Upon precipitation with phenol-chloroform-isoamyl alcohol (Sigma-Aldrich), RNA was retro-transcribed to cDNA using the RevertAid First-Strand cDNA Synthesis kit (Thermo Scientific) and analyzed by RT-qPCR with PyroTaq Eva-Green reagents in a QuantStudio 3.0 0.1 mL thermocycler (Applied Biosystems). The sequence of the primer pairs used for RT-qPCR is shown in Supplementary Table 1.

## Chromatin immunoprecipitation (ChIP) assays

H3K4me3, H3K27ac, CTCF, and H3K27me3 ChIPs were performed as described in[79] with minor modifications. The following antibodies were used: ChIP antibodies: α-Kdm1a (ab17721, Abcam); α-H3K4me3 (07-473, Millipore); α-H3K4me1 (ab8895, Abcam); α-H3K27ac (ab4729, Abcam); α-CTCF (07-729, Millipore); and α-H3K27me3, (07-449, Millipore). Briefly, 3 μg of antibody were incubated with Dynabeads coated to Protein G (Invitrogen) in RIPA-150 buffer (50 mM Tris-HCl, 150 mM NaCl, 1 mM, EDTA, 0.1% SDS, 1% Triton X-100, and 0.1% sodium deoxycholate; pH 8) for 6-18 h at 4 °C in agitation. Nuclei from hippocampal cells were extracted in a dounce homogenizer as explained for the FANS method. Samples were fixed with 1% formaldehyde (Sigma-Aldrich) for 10 min at room temperature followed by 0.1 M glycine for 5 min to stop the fixation. Nuclei were centrifuged at 1,5 G, and the pellet was resuspended in 100 μL of SDS lysis buffer (1% SDS, 10 mM EDTA, 50 mM Tris; pH 8) prior to its sonication in a Bioruptor Pico (Diagenode) for 12 cycles of 15 s on/30 s off. After 6 min of centrifugation at 17000 x g, the supernatant containing isolated chromatin was collected. Samples were diluted in 900 μL of ChIP Dilution buffer (0.01% SDS, 1.1% Triton X-100, 1.2 mM EDTA, 16.7 mM Tris-HCl, 167 mM NaCl; pH 8) and incubated overnight at 4 °C with the antibody-Dynabeads mix. Beads were thoroughly rinsed at 4 °C as follows: two washes in RIPA-150 buffer, three washes in RIPA-500 buffer (50 mM Tris-HCl, 500 mM NaCl, 1 mM, EDTA, 0.1% SDS, 1% Triton X-100, 0.1% sodium deoxycholate; pH 8), two washes in RIPA LiCl buffer (50 mM Tris-HCl, 1 mM EDTA, 1% NP-40, 0.7%, sodium deosycholate, 500 mM LiCl$_2$), and two final washes in TE buffer (10 mM Tris-HCl, pH 8.0, 1 mM EDTA; pH 8.0), five minutes each. Samples were resuspended in 200 μl Elution buffer (10 mM Tris-HCl, 5 mM EDTA, 300 mM NaCl, 0.5% SDS; pH 8) and treated with 1 μl of 10 mg/mL RNase A (Fermentas) overnight at 65 °C in agitation for crosslink reversion. Samples were treated with 3 μl of 20 mg/mL Proteinase K (Thermo Scientific) for 2 hours at 55 °C in agitation. Last, DNA precipitation was performed by phenol-chloroform-isoamyl alcohol (Sigma-Aldrich). For Kdm1a ChIP, some modification were applied as in ref. 80. In brief, samples required a stronger fixation treatment in 1% PFA for 30 min at 37 °C. To fragment these highly fixed samples, 43 cycles of sonication (30 s On, 30 s Off) were applied in a Bioruptor Pico (Diagenode). SDS lysis buffer was modified to RIPA buffer (0.1% SDS, 1% IGEPAL, and 0.5% sodium deoxycholate).

## RNA-seq analysis

Hippocampi were isolated and RNA extracted as described in RT-qPCR assays. Three mice were used per genotype (Kdm1a-ifKO and their control littermates). Poly-A libraries were generated for each independent sample and single-end sequenced on a HiSeq 2500 apparatus from Illumina. Adapters were trimmed using cutadapt v1.18 selecting reads longer than 25 bp. HISAT2[81] (v2.1.0) was used for reads alignment to the mouse genome (GRCm38.89, mm10) and reads with mapq > 30

and that mapped to nuclear chromosomes were quantified using HTSeq v0.11.1[82]. Data processing was performed with custom R scripts (v3.5.1, 2018), Samtools v1.9[83], Bedtools v2.27.1[84], and DeepTools v3.5.0[85]. Whole genome alignments were normalized to 10× RPM (read per 10 million sequenced reads). Differential expression analysis was conducted using DESeq2[86] (v1.18.1). For gene upregulation, significant differences were considered if p adj <0.05 and log$_2$FC >1 or 0.5, depending on the analysis (specified in the text). For gene down-regulation, all genes with p adj <0.05 were considered, given the low number of downregulated genes. *Biological process* and *Molecular function* gene ontology analysis, and *Chromosome location analysis* were performed using WebGestalt (WEB-based GEne SeT AnaLysis Toolkit[87]; www.webgestalt.org), using over-representation analysis (ORA) as the enrichment method. The enrichment for binding motifs was performed using Enrichr, a gene set enrichment analysis tool based on the Encyclopedia of DNA Elements (ENCODE) and ChIP-x Enrichment Analysis (ChEA) databases[88]. RNA-seq counts were normalized by reads per million. To classify genes as expressed or not expressed in neurons, we used the nuclear RNA-seq data previously generated in our lab[26]. Briefly, after removing adapters using Trim-Galore v0.6.4_dev (https://www.bioinformatics.babraham.ac.uk/projects/trim_galore/), we aligned to mm10 using STAR[89], and obtained the counts using Rsubread (v. 2.4.3)[90]. Finally, we normalized the reads using Transcripts per million and selected those with an expression bigger than 5. From this list we randomly selected a similar number of regions tested.

## ChIP-seq analysis

The hippocampal formation was extracted, and chromatin immuno-precipitation was performed as aforementioned. For H3K4me3, H3K27me3, and H3K27ac ChIP-seq, two or three independent samples were processed per genotype (Kdm1a-ifKO, and their control littermates), the pool of two mice per sample to H3K4me3, one mice per sample to H3K27me3, H3K27ac or CTCF. For Kdm1a ChIP-seq, two WT mice were pooled in one sample. 50 bp single-end sequencing was performed using a HiSeq 2500 sequencer. Adapters were trimmed using TrimGalore v 0.6.4_dev (https://www.bioinformatics.babraham.ac.uk/projects/trim_galore/), and the reads obtained were mapped to the mouse genome (mm10) with Bowtie2 (v2.3.4.3)[91]. Only those reads with a mapq > 30 were selected. Data was processed using Samtools v1.10[83], Deeptools v 3.5.0[85], and Bedtools v2.25.0[84]. For BigWigs, ChIP-seq reads were normalized by reads per genomic content with Deeptools[85], except in the case of H3K4me3 and H3K27me3 where the scalefactor given by Deseq2 was used. Also, note that the H3K27me3 regions used for scalefactor normalization came from the peak calling conducted with the datasets generated in[92]. Peakcalling was performed with macs2 (v2.1.1.20160309)[93], and differential binding analysis was performed with Diffbind (v 2.10.0)[94]. Annotation was performed using ChipPeakAnno (v3.24.2)[95] and Biomart (v2.46.3)[96]. In the case of H3K4me1, the initial PCA identified two outlier samples (one per genotype) that were confirmed by observation of the individual profiles and was removed in further analyses. The differential enrichment analysis did not detect significant differences between genotypes for the H3K4me1 ChIP-seq. In Supplementary Fig. 3F, to identify the regions with high content of H3K27me3, we divided the signal in the dataset GSE56810[92] using *plotHeatmap* from *Deeptols* and the setting "–kmeans 2", and selected the cluster with the strongest H3K27me3 signal. For Kdm1a peaks, the Irreproducibility Discovery Rate (IDR) algorithm was used; we selected those peaks that presented a scaled IDR value larger than 955. For violin plots, we used homer (v4.11.1)[97] to select the tags of the regions of interest, and the plot was performed using ggplot2 (v3.3.3)[98]. In the analysis of H3K27me3 data from human samples, wigs were transformed to bigwis using wigtobigwig from ENCODE and the profiles were performed with Deeptools.

## ChIA-PET analysis

Chromatin Interaction Analysis by Paired-End Tag Sequencing (ChIA-PET) using a CTCF-specific antibody (Abclonal Cat # ab70303) was performed as described in the in situ ChIA-PET protocol[45] on approximately 10 million nuclei of hippocampal excitatory neurons sorted by FANS as described above. The ChIA-PET library was sequenced with 150 bps long pair-end reads and was processed by the ChIA-PIPE pipeline[99] using mm10 as a reference genome and default parameters. The resulting *hic file for 2D contact maps and *bigwig file for protein binding coverage were used for visualizing on Juicebox (Durand et al., 2016) Within ChIA-PIPE, peaks were called via MACS2 without input control and only those with the maximum coverage greater than 50 (which is twice the median of all peaks) were kept by pybedgraph (Zhang et al., 2020), resulting in 34,885 confident peaks. Using these peaks, the loops reported in *bedpe format with 7th column denoting PET counts are further filtered to have PET count greater than or equal to 3 and to have both left and right anchors to overlap with at least one peak. These 27,786 loops were the subject of ulterior analyses. In Fig. 3G, the CTCF-ChIA-PET contact map in ".hic" format was normalized with the 'Coverage (Sqrt)' option visualized in Juicebox (Aidenlab). In Fig. 3H, the CTCF PET counts indicate the summation of ChIA-PET counts at CTCF loops that encapsulate 100% of the gene body (the longest loop, if there are multiple isoforms). We did not impose any rules on distance from the gene. To calculate the loop-span the same encapsulation loops were taken and the average distance between the CTCF peaks that encapsulate the genes was measured. For Fig. 4D, we used *pairtobed* from bedtools for selecting those CTCF that were in the upDEGs with a $\log_2FC > 0.5$ and padj <0.05. We found that 1470 loops are associated with regulatory regions (±10.000 bp upstream and downstream of the genes and into the genebody) of upregulated genes in Kdm1a-ifKO neurons. CTCF-loops of upDEGs are associated through CTCF with other genes (aka CTCF-associated genes). To analyze the epigenetic and transcriptional behavior of the loops associated with the upDEGs, the signal density was plotted to different epigenetic profiles obtained in this study. For them we called the profiles matching both CTCF positions and the metagene from the RNA-seq and Chip-seq profiles (H3K27ac, H3K27me3, H3K4me3, CTCF, and Kdm1a) on both sides of the CTCF connections involving the upDEGs. All of this was performed using R and bash scripts and Bedtools. For Supplementary Fig. 6A, we selected either the left or right CTCF sites for each loop and looked for H3K27me3 enrichment (GSE56810) using Deeptools with the parameter *–kmeans*. In the clusters that displayed H3K27me3 enrichment, we plotted the H3K27ac and LSD1 signal density on the opposite side of the loop. Finally, we obtained the number of H3K27ac-rich sites using the *–kmeans* parameter.

## Hi-C analysis

The chromatin compartment mapping shown in Fig. 3C and Fig. 4B is based on the neuron-specific Hi–C data from adult mouse hippocampus previously generated in our laboratory[26]. Sub-compartment information comes from Supplementary Data 5 in ref. 40. That article applied k-based sub-compartment mapping to reanalyze the neuron-specific Hi–C data from adult mouse hippocampus[26]. Note that the compartment mapping presented in Fig. 3C and Fig. 4B was conducted at 25 Kb resolution, whereas Chandrasekaran and colleagues used 250 Kb resolution. The difference in resolution resulted in an enlargement of the B compartment in the analysis by Chandrasekaran et al. Source data are provided with this paper.

## Primary cultures and lentiviral infection

Primary hippocampal cultures were prepared from Kdm1a[f/f] embryos. Their hippocampi were dissected, pooled together, and dissociated for neuronal extraction[100]. Cells were plated in 24-well plates at $0.11 \times 10^6$ neurons/well. After 24 h, the primary cultures were infected with the indicated viruses (day in vitro 1, DIV1). Rescue experiments were performed with a short variant of human KDM1A, without the alternative exons 2a and 8a. To test the relevance of the IDR, the same protein but lacking the amino terminal IDR was expressed.

## Phase separation analyses

For fluorescent recovery after photobleaching (FRAP) experiments and droplet formation in vivo 40,000 HEK293T cells (CRL-3216™ from ATTC) growing on 35 mm dish with 20 mm micro-well and 1.5 glass-like polymer coverslip were transfected using encoding hKDM1a (transcript variant 2, NM_015013.4) fused to eGFP. Flag-HA tag was replaced by the GFP fluorescent protein in pOZ-FH-hKdm1a expression vector donated by the Yang Shi laboratory. FRAP was performed on a ZEISS LSM880 AiryScan Elyra S1 SR Confocal microscopy. The spot was bleached using zoom bleach with 15 interactions at nominal 100% laser transmission (405 nm diode). Images were acquired using a ×63 oil-immersion objective, 3× optical zoom and 1024×1024-pixel resolution. We took 5 images as controls before bleaching and then 1 image every 50 cycles of 10 second immediately after bleaching. Time-lapse images were acquired at 2 second intervals. For 1,6 hexanediol treatment HEK293T cells were coated with poly-D-lysine on glass dishes and transfected with 0.5 μg of GFP-hKdm1a. 1.6 hexanediol was added to the culture medium to a final concentration of 2.5%, washed with PBS, and fixed with 4% PFA after 5 minutes. As control, we used GFP-hKDM1A transfected cells not treated with 1,6 hexanediol. Confocal images were obtained on an Olympus FV1200 Confocal using a x63 oil-immersion objective at a 5x optical zoom and a 1024×1024-pixel resolution. Raw images were analysed using Fiji software for subsequence quantification.

## Human samples analysis

RNA expression data from control postmortem brain samples were obtained from the NCBI Gene Expression Omnibus (GEO) database (GSE33000, GSE15222 and GSE48350) and analyzed using R software (http://www.R-project.org). Low-represented extreme values (below 50 and above 90 years) were not considered for the calculation of age-RNA correlations to increase the reliability of the results. Only control samples were used for the analysis. Z-score normalization was applied prior to merging. Scores were calculated as the average of the scaled expression values (range 0 and 1) of the corresponding genes.

## Statistical methods

All statistical analyses were conducted using SigmaPlot v12.5 (Systat Software, Inc., San Jose CA), GraphPad Prism v7.04 (GraphPad Software, La Jolla CA), R v3.6, and RStudio v1.1.447. The statistical test used for each analysis is specified in the corresponding figure legend. When comparing two groups, a normality test was first performed using the Shapiro-Wilk test. If samples followed a normal distribution, t-test was conducted; otherwise, Mann–Whitney U statistic was used. Box plots are interpreted as follows: first and third quartiles are indicated with the bottom and top lines of the box, respectively; the median and mean of the samples are indicated with the solid and dashed lines within the box, respectively; the whiskers below and above the box indicate the 10th and 90th percentiles, respectively; raw data are presented as dots on top of the plot. Barplots represent means ± s.e.m. In the violin plot, Wilcoxon rank sum test was performed. For the cumulative plot, genes were first ordered by location in the genome, the distance between consecutive TSSs were measured, and they were ordered from smaller to largest in the X axes. Y axes represents the cumulative frequency of each of the distances. P values were considered significant when α was lower than 0.05. To calculate the statistical significance of the overlap between two groups of genes, we used the online tool available at nemates.org. A representation factor > 1 indicates more overlap than expected by chance, while a representation factor <1 indicates less overlap than

expected. The accompanying statistic is based on exact hypergeometric probability.

## Inclusion and diversity statement
We support inclusive, diverse, and equitable conduct of research.

## Reporting summary
Further information on research design is available in the Nature Portfolio Reporting Summary linked to this article.

## Data availability
The genomic data sets generated in this study can be accessed at the GEO public repository using the accession number GSE236182. In addition, we used several previously published datasets: GSE123652: Kdm1a ChIP-seq from adult PFC[47]; GSE133018: CBP and H3K27ac ChIP-seq from adult mouse hippocampus[80]; GSE56810: H3K27me3 ChIP-seq from adult mouse hippocampus[92]; GSE125068: RNAPII ChIP-seq from adult mouse hippocampus, and Hi-C, ATAC-seq and nuclear RNA-seq from sorted mouse hippocampal principal neurons[26]; GSM2393589: H2AK119ub1 ChIP-seq from mouse NPCs[101]; GSM1479208: H2AK119ub1 ChIP-seq from mouse brain[102]; GSM918727: CTCF ChIP-seq from adult mouse cortex[43]; GSM1917302: Suz12, and GSM1917302: Ezh2 ChIP-seq from NPCs[103]. Sub-compartment information come from Supplementary Data 5 in[40], which applies k-based sub-compartment mapping to the neuron-specific Hi−C data from adult mouse hippocampus was used in the analyses presented in Fig. 3C and Fig. 4B[26]. Human RNA expression data from control postmortem brain samples were obtained from GSE33000, GSE15222 and GSE48350. Source data are provided with this paper.

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

## Acknowledgements

We thank Ana Pombo and Victor Corces for the critical reading of the manuscript. We thank the personnel of the Mouse facility and the Microscopy and Omics core services at the Instituto de Neurociencias for their collaboration and R. Olivares for technical assistance. S.N. is the recipient of a fellowship from the Spanish Ministry of Science and Innovation (MICINN). A.M.M-G. was the recipient of a *Bio4Med (Biology for Medicine): International Doctoral Studies on Biological Bases of Human Diseases* fellowship given by the Polish Ministry of Science and Higher Education (MNiSW). J.P-L. is the recipient of a Margarita Salas contract from the Spanish Ministry of Universities funded by Next Generation EU. A.B. research is supported by grants #202003 from Fundació la Marató de TV3, PID2020-118169RB-100 from AEI co-financed by ERDF, PCI2021-122087-2A/ERA-Net NEURON NDD-243, from AEI, PROMETEO/2020/007 from the Generalitat Valenciana, and FCAIXA HR22-00394 from Fundación LaCaixa. M.K. is supported by the National Institutes of Health (K99-HG011542; R00-HG011542). J.V.S-M. research is supported by the Spanish Ministry of Science and Innovation (PID-2019-111240RA-I00), and the CSIC Interdisciplinary Thematic Platform (PTI) NEURO-AGING+. Both A.B. and Y.R. were supported by the RGP0039/2017 from the Human Frontiers Science Program Organization (HFSPO). The Instituto de Neurociencias is a "Centre of Excellence Severo Ochoa" (CEX2021-001165-S).

## Author contributions

Conceptualization, B.dB., S.N., A.M.M-G. and A.B.; Methodology, B.dB., S.N., A.M.M-G., J.P-L., M.K., R.M-V., and C.R.; Software, S.N., M.K.,J.V.S-M. and R.M-V.; Investigation, B.dB., S.N., A.M.M-G., J.P-L., M.K. and R.M-V., Data Curation and Visualization, B.dB., S.N., M.K. and J.V.S-M; Writing – Original Draft, B.dB. and A.B.; Supervision, A.B., Y.R., and B.dB.; Funding Acquisition, A.B.

## Competing interests

The authors declare no competing interests.
