## [Peer Review File · Nature Communications]

Kdm1a safeguards the topological boundaries of PRC2-repressed genes and prevents aging-related euchromatinization in neuronsREVIEWER COMMENTS

Reviewer #1 (Remarks to the Author):

This study by del Blanco, Ninerola and colleagues presents an enormous amount of work, essentially a multi-omic work up of a molecular novel phenotype resulting from neuron-specific induced ablation of Kdm1a histone demethylase,

The paper nicely shows by cell-type specific chromatin and RNA-seq assays, and superresolution microscopy, and even homology-based analysis of age-related human brain neurogenomics, that KDM1a and Kdm1a-mediated H3K4 methylation in neurons is essential for transcriptional repression of non-neuronal genes mostly positioned in A-compartment (presumably open) chromatin engaged in complex loop interactions between proximal H3k27me3-defined (repressed) sequences at the site of the regulated genes with more distal , H3K27ac-defined (active enhancers?) sequences.

Overall I applaud the Authors to this excellent paper that will become a landmark study for the field.

I have a few comments:

1) if I read the paper correctly, then most of their Kdm1a-regulated genes are repressed genes located in the A-compartment(Which is broadly linked to transcriptional activation/facilitation etc). this is fine. However, I feel the authors tend to 'replace' in their title and abstract and portions of the A-compartment concept (which is based on PC of Hi-C chromosome conformation mapping) with the conceptually more attractive term "euchromatin" (which is, strictly speaking, not what the authors measured . I think the Discussion section is perfectly suited to discuss the concept of euchromatin and euchromatinization in the context of their Kdm1a mutant studies, but I feel the term is misleading if used in the abstract, title, results .

2) It is very interesting that the Authors discovered KDM1a-regulated H3k27me3-rich domains in the A compartment. A/B compartments can be further subdivided into A1, A2... etc (Admittedly, the tools to do this, K-means clustering, is somewhat arbitrary). Have the authors checked if their KDM1a regulated genes fall within a specific A compartment subtype?

3) Figure 4B: Kdm1a chip shows enrichment in A-compartment. But is it enriched in the subset of A-compartment blocks that harbor one or more Kdm1a-regulated genes?

4) The Authors use site-specific chromosome conformation capture and describe loop-remodeling at the PPP1R14D/Spint1pair and Ino80 locis. This is a conceptually extremely important observation (contributing to spreading of active marks into repressed genes). Could the authors comment on whether they have indirect evidence for or against such type of mechanism for the collective group of Kdm1a-regulated genes?

Reviewer #2 (Remarks to the Author):

The study by del Blanco et al. investigates epigenetic and transcriptional consequences of ablating Kdm1a, a H3K4 demethylase implicated in intellectual disability, in adult excitatory neurons in mice. Combining in vivo multi-omics, including RNA-seq, ChIP-seq, FANS-ChIA-PET and 4C-seq, advanced microscopy and in vitro analyses, they show that neuronal loss of Kdm1a leads to up-regulation of non-neuronal genes that are normally repressed and enriched in H3K27me3, the major target of the Polycomb repressive complex 2 (PRC2). Gene upregulation upon Kdm1a suppression is associated with their euchromatinization (i.e. increased H3K4me3, increased H3K27ac, reduced H3K37me3). The data

indicate that the underlying mechanism involves disrupted 3D organization of small CTCF-dependent chromatin loops/domains containing the up-regulated non-neuronal genes, which are embedded in broader active domains. The data further indicate that Kdm1a displays liquid-liquid phase separation (LLPS) properties, and that suppression of Kdm1a-mediated LLPS in neurons contribute to euchromatinization of the small heterochromatinized domains containing kdm1a-sensitive genes. Finally, the authors provide some evidence that this mechanism is aging-related and conserved in humans. Overall, this is an original study, which is well conducted and provides novel insights into neuronal function of Kdm1a, a major epigenetic regulator, whose dysregulation is relevant to various neurological conditions, including intellectual disability, neurodegenerative disorders and possibly brain aging. Additionally, the data represent valuable resources to the community. Nonetheless, I have few concerns that need to be addressed to improve the manuscript.

1- The main targets of Kdm1a are H3K4me2/1; H3K4me3 is not a direct target of Kdm1a. H3K4me1 and H3K4me3 ChIP-seq data have been generated in the study; however, H3K4me2 ChIP-seq data were not produced. Moreover, the analysis of H3K4me1 ChIP-seq data is limited: only one figure is shown (in Fig. 2B), and it is focused on promoter regions. However, H3K4me1 is enriched at enhancers. Therefore, it might be wise to analyze enhancer regulation (and integrate H3K4me1 enhancer analysis with additional ChIP-seq data generated in the project, including Kdm1a, H3K27me3, H3K27ac... ChIP-seq). Indeed, loss of clear boundaries between repressed regions containing upDEGs and active domains in which they are imbedded could result, at least in part, from altered regulation of enhancers. Kdm1a has been shown to suppress transcription by silencing enhancers: are H3K4me1/2 increased at enhancers in Kdm1a ko cells, which could contribute to upDEGs?

2- What is the overlap/correlation between RNA-seq DEGs and ChIPseq DERs upon Kdm1a neuronal depletion?

3- The authors show that promoters of upDEGs are enriched in H3K27me3 and also, to some extent in H3K4me3, which suggest that Kdm1a-sensitive genes are bivalent genes. Could the authors specifically analyze this? What is the proportion of upDEGs which are bivalent, that the presence of H3K4me3 at upDEG correlate with Kdm1a binding?

4- What is the impact of Kdm1a loss in neurons on activity-regulated genes? The authors have treated mice with KA and show that Fos is increased upon KA treatment both in Kdm1a KO mice and control mice. What about additional IEGs, including Egr1, which is strongly dependent on SRF. It might be interesting to investigate such IEGs, since corepressor complex comprising Kdm1a was implicated in the repression of SRF target genes.

5- What is the expression level of upDEG-associated genes (they are expected to be repressed if H3K27me3 is increased, as shown in Fig.4D)? RNA-seq data using rpkm could be used to show expression results, it is more quantitative than metaprofiles. Additionally, how to explain that generally Kdm1a loss has no impact on transcription of active genes? It could be discussed

6- Fig6C-F, analysis of positive and negative control proteins (with and devoid of LLPS properties, respectively) would strengthen the results

7- P17, line 398, "this results is consistent with a recent study showing that the euchromatinization of H3K27me3-repressed genes is a key feature of aging tissues (Yang et al. 2023b)". In this paper focusing on mouse liver, the authors show that aging is associated with both derepression of PRC2 target genes (as found in the current study) and global increase of H3K27me3 at age-domains/LADs, which are broad gene-poor domains. It would be interesting to examine whether loss of Kdm1a also leads to epigenetic changes at those broad domains, which would provide further evidence that Kdm1a loss mimicks aging.

Minor

- Fig.1D, other histone demethylases than Kdm1a or histone methyltransferase are not changed upon Kdm1a inactivation, as shown by RNA-seq analysis. However, this does not exclude changes at protein level/activity. The conclusion should be tone down/discussed
- Fig2E comes before Fig2D, this needs to be corrected
- FIS3E: Cbx1,3 and 5 are not PRC1 subunits; Cbx2/4/6/7/8 are PRC1 subunits, their profiles should be shown instead of that of Cbx1/3/5.
- The two graphs in Fig3A are unclear, the legend is not enough explicit (what is 8B3.3, 4D3...?).
- In Fig3B, the difference does not appear huge, what is the statistics?
- Fig. 3D aims to show that upDEGs are organized in clusters. Could this be quantified, relative to gene density for instance?
- Fig.3G, ChiAPET data. The legend of Fig3G could be more detailed, minimal information is provided.
- Fig3H, how is loop span calculated?
- FigS4D, what is the distribution for all genes? To suggest that upDEGs are enriched in CTCF regulated genes, a reference for comparison is needed.
- The statistics shown in FigS4F are unclear
- Fig.4C-D, is it possible to quantify the proportion of upDEGs associated with loops, presenting one anchor in an active region bound by Kdm1a? and p12 & FigS6A: "some CTCF loops presented H3K27me3 at one CTCF anchor and H3K27ac and KDM1a at the opposite" Could it be quantified? What about the other loops with upDEGs?
- In Fig.S6BC, ChIPseq profiles in both WT and Kdm1a ko should be shown when possible
- Fig.4D, it seems that H3K27me3 is increased at CTCF peak associated with up DEG-associated genes, and that H3K4me3 and transcription are also low (see 'iv')
- Also strikingly RNA peaks (see 'iv') are greater at TES than TSS, why is it so?
- Fig4D. the representation (i), (ii)....in this figure is unclear: the CTCF peak directly linked to upDEGs is centered to TSS (see i) and it seems it does not match with the TSS of upDEGs shown in (ii). Similarly, the CTCF peak directly linked to upDEG-associated genes is centered to TSS (iii), but it does not seem to match to what is shown in (iv). Peak intensities are different. It is unclear why CTCF peaks are centered to TSS.
- Fig.4E, it would be wise to also show mRNA levels of Spint1, Vps18, Ino80 and additional genes in the region, in Kdm1a ko and control conditions. Also H3K4me3 and H3K4me1 profiles could be included. Generally, profiles of both control and Kdm1 ko could be shown.
- Fig4F and S6G (4Cseq data): changes in interaction strength in Kdm1a vs control at Spint1 is unclear. The representation/figure legend could be improved. Statistics for 4Cseq data are not shown, though it is said that differences are significant. It is difficult to know the extent of the changes. A representation showing 4Cseq signals would be more informative.
- Moreover, data related to Ppp1r14d are not shown, only data for Spint1 are shown.
- Generally, legend figures could be improved, particularly providing more details.

Reviewer #3 (Remarks to the Author):

The manuscript by del Blanco et al. presents a nice and rather thorough investigation of the role of Lsd1 (Kdm1a) in neurons of the adult mouse forebrain, based on epigenomic, transcriptomic, and genomic architectural analysis of conditional knockout mice. They also deftly utilize publicly available multi-omics data to support their model, which proposes that Lsd1, by virtue of its N-terminal IDR, maintains the repression of nonneuronal genes in neurons through direct and indirect collaboration with the PRC2 complex and CTCF, respectively.

As the authors note, a few studies have already addressed the potential function of Lsd1 in neurons; however, the manuscript does not sufficiently integrate their new results with previously published

data on this topic. At the very least, additional explanation is needed to address any discrepancies as well as the relevance of unexplored aspects, some of which are highlighted below:

- The difference in phenotype and hippocampal gene expression changes in the conditional *Lsd1* knockout mice in this study vs. earlier work (Christopher et al., 2017) that employed a Cagg-Cre tamoxifen-inducible system is striking and rather surprising. Is there any overlap in the RNA-seq changes detected in the two studies?
- Two different neuronal isoforms of *Lsd1* have been described, each with altered histone specificity (Laurent et al., 2015; Wang et al., 2015). These isoforms are presumably absent in the conditional knockout. Accordingly, are any global changes in H4K20me1/2 or H3K9me2 expected that might contribute to the observed gene expression changes?

It is interesting that few of the de-repressed genes in the conditional *Lsd1* knockout neurons actually recruit *Lsd1*. The authors note that ~60% of these dysregulated genes harbor proximal CTCF sites, but what do the authors think is going on with the remaining ~40%? Do these have more distal CTCF sites that might still be functionally significant, or do the authors suspect that other collaborating factors may be involved?

This sentence in the Supplementary Figure S8 caption is confusing and seemingly inaccurate: "RT-qPCR assays demonstrate that human *Kdm1a* bearing mutations (K661A and K598E) that abolish its H3K4me2 demethylase catalytic activity do not prevent upDEG activation as efficiently as the h*Kdm1a* protein lacking the IDR." The *LSD1* catalytic mutants largely prevent de-repression, unlike the IDR deletion mutant, which is presumably the point.

Finally, regarding the suggestion that de-repression of *Lsd1*- regulated genes might be a feature of brain aging, how do the authors envision the involvement of *Lsd1* itself? Is there any evidence that levels of *Lsd1* (or of PRC2 components) change in neurons with age?

These outstanding questions/issues notwithstanding, the manuscript of del Blanco et al. is instructive and should add an important increment to our understanding of the functional significance of *LSD1* in mature neurons.

Point-by-point response to the reviewers' comments

We thank the Editor for giving us the opportunity to improve the manuscript and the three Reviewers for their thoughtful and constructive comments. We conducted new analyses and experiments according to the Reviewers comments. These new results are presented in new or revised panels in **Figures 2 and 4, Supplementary Figures S1, S2, S3, S4 and S6**, and in this letter, and in the new **Table S3**. In addition, we revised the text and figures to incorporate the changes suggested by the Reviewers and corrected a few minor mistakes. We believe that with these changes and additions we have effectively addressed the Reviewers' comments and the manuscript is significantly improved. Our point-by-point response to all the issues identified during the review process is below:

Reviewer #1 (Remarks to the Author):

This study by del Blanco, Ninerola and colleagues presents an enormous amount of work, essentially a multi-omic work up of a molecular novel phenotype resulting from neuron-specific induced ablation of Kdm1a histone demethylase,

The paper nicely shows by cell-type specific chromatin and RNA-seq assays, and superresolution microscopy, and even homology-based analysis of age-related human brain neurogenomics, that Kdm1a and Kdm1a-mediated H3K4 methylation in neurons is essential for transcriptional repression of non-neuronal genes mostly positioned in A-compartment (presumably open) chromatin engaged in complex loop interactions between proximal H3k27me3-defined (repressed) sequences at the site of the regulated genes with more distal , H3K27ac-defined (active enhancers?) sequences.

Overall I applaud the Authors to this excellent paper that will become a landmark study for the field.

We thank Reviewer #1 for the very positive appreciation of our work. We also believe that our work contributes to a better understanding of the regulation of chromatin boundaries and transcription in adult brain neurons and thereby has the potential to become a landmark study in the field.

I have a few comments:

1) if I read the paper correctly, then most of their Kdm1a-regulated genes are repressed genes located in the A-compartment(Which is broadly linked to transcriptional activation/facilitation etc). this is fine. However, I feel the authors tend to 'replace' in their title and abstract and portions of the A-compartment concept (which is based on PC of Hi-C chromosome conformation mapping) with the conceptually more attractive term "euchromatin" (which is, strictly speaking, not what the authors measured. I think the Discussion section is perfectly suited to discuss the concept of euchromatin and euchromatinization in the context of their Kdm1a mutant studies, but I feel the term is misleading if used in the abstract, title, results .

We appreciate the Reviewer's comment, and we have made the necessary revisions in response to his/her feedback. Specifically, we have corrected instances where we mistakenly used the term "euchromatin" when referring to compartment A. These revisions have been applied throughout the text, including the sentence in the abstract.

However, in the case of the title, we believe that the term "*euchromatinization*" is still appropriate. This choice is supported by our findings, which demonstrate a significant shift in the epigenetic profile of affected genes. We observe a transition from an enrichment of the repressive mark H3K27me3, as well as the absence of H3K27ac and H3K4me3, to a permissive state characterized by the enrichment of H3K4me3 and H3K27ac at the promoter and depletion of H3K27me3. These epigenetic changes are indicative of euchromatin formation. Therefore, we maintain our preference for the original title. As an alternative, we could consider using the term "*ectopic gene activation*" but we strongly favor retaining "*euchromatinization*" in the title.

2) *It is very interesting that the Authors discovered Kdm1a-regulated H3k27me3-rich domains in the A compartment. A/B compartments can be further subdivided into A1, A2... etc (Admittedly, the tools to do this, K-means clustering, is somewhat arbitrary). Have the authors checked if their Kdm1a regulated genes fall within a specific A compartment subtype?*

This is an excellent suggestion to extend the analysis of upDEGs location. To carry out the new analysis, we used the Hi-C sub-compartment annotation produced by Chandrasekaran et al., (*Nat Comm*, 2021). This article applied k-based sub-compartment mapping to the neuron-specific Hi-C data from adult mouse hippocampus generated in our laboratory (Fernandez-Albert et al., *Nat Neurosci*, 2019), which is the same dataset used in the analyses presented in **Fig. 3C** and **Fig. 4B**. Importantly, we should note that the compartment mapping presented in **Fig. 3C** and **Fig. 4B** was conducted at 25 Kb resolution, whereas Chandrasekaran and colleagues used 250 Kb resolution. This has caused significant differences in the distribution of the A and B compartments, resulting, comparatively, in an enlargement of the B compartment in the analysis by Chandrasekaran et al.

The analysis suggested by Reviewer #1 demonstrated that *Kdm1a*-regulated genes are distributed between the sub-compartments A1 and A2 in adult excitatory neurons. We did not detect any clear preference for a sub-compartment or the other. This information has been added in page 10 of the main text, page 14 in *Supplementary Materials* and new **Supp. Fig. S4B**. Interestingly, the new analysis also revealed that 15% of the upDEGs that mapped into the A compartment at 25Kb resolution now mapped into the B1 compartment defined in the 250Kb resolution study. This result suggests that these genes locate near the boundary between A/B compartments.

Figure legend: The sector plots represent the percentages of upDEGs distributed in compartments A and B (left) as shown in Fig. 3C, or in subcompartments A1, A2, B1 and B2 (right) as defined by Chandrasekaran et al., 2021. B1 (←A) corresponds to the group of genes annotated into the A compartment at 25Kb resolution that mapped into the B1 compartment at 250Kb resolution.

3) Figure 4B: Kdm1a chip shows enrichment in A-compartment. But is it enriched in the subset of A-compartment blocks that harbor one or more Kdm1a-regulated genes?

We obtained similar results with the distribution of Kdm1a peaks between sub-compartments. No apparent bias in the distribution between the A1 and A2 compartments were observed. Moreover, we also found that many peaks at the A compartment at 25Kb resolution remapped into the B1 compartments when the lower resolution was used, suggesting that these peaks locate in the proximity of regions separating the A and B compartments.

Figure legend: The sector plots represent the percentages of Kdm1a peaks distributed in sub-compartments A1, A2, B1 and B2, as defined by Chandrasekaran et al.

4) The Authors use site-specific chromosome conformation capture and describe loop-remodeling at the PPP1R14D/Spint1pair and Ino80 locis. This is a conceptually extremely important observation (contributing to spreading of active marks into repressed genes). Could the authors comment on whether they have indirect evidence for or against such a type of mechanism for the collective group of Kdm1a-regulated genes?

We do not have such evidence for the collective group of Kdm1a-regulated genes. However, Fig. 4D shows that the chromatin remodeling leading to spreading of active marks into repressed genes can be detected in the global analysis with the group of upDEGs, suggesting that loop remodeling may be a more global phenomenon.

Reviewer #2 (Remarks to the Author):

The study by del Blanco et al. investigates epigenetic and transcriptional consequences of ablating Kdm1a, a H3K4 demethylase implicated in intellectual disability, in adult excitatory neurons in mice. Combining in vivo multi-omics, including RNA-seq, ChIP-seq, FANS-ChIA-PET and 4C-seq, advanced microscopy and in vitro analyses, they show that neuronal loss of Kdm1a leads to up-regulation of non-neuronal genes that are normally repressed and enriched in H3K27me3, the major target of the Polycomb repressive complex 2 (PRC2). Gene upregulation upon Kdm1a suppression is associated with their euchromatinization (i.e. increased H3K4me3, increased H3K27ac, reduced H3K37me3). The data indicate that the underlying mechanism involves disrupted 3D organization of small CTCF-dependent chromatin loops/domains containing the up-regulated non-neuronal genes, which are embedded in broader active domains. The data further indicate that Kdm1a displays liquid-liquid phase separation (LLPS) properties, and that suppression of Kdm1a-mediated LLPS in neurons contribute to euchromatinization of the small heterochromatinized domains containing kdm1a-sensitive genes. Finally, the authors provide some evidence that this mechanism is aging-related and conserved in humans. Overall, this is an original study, which is well conducted and provides novel insights into neuronal function of Kdm1a, a major epigenetic regulator, whose dysregulation is relevant to various neurological conditions, including intellectual disability, neurodegenerative disorders and possibly brain aging. Additionally, the data represent valuable resources to the community. Nonetheless, I have few concerns that need to be addressed to improve the manuscript.

We thank the Reviewer for the positive comments and general appreciation of the study. We respond point-by-point to Reviewer #2 questions as follow:

1- The main targets of Kdm1a are H3K4me2/1; H3K4me3 is not a direct target of Kdm1a. H3K4me1 and H3K4me3 ChIP-seq data have been generated in the study; however, H3K4me2 ChIP-seq data were not produced. Moreover, the analysis of H3K4me1 ChIP-seq data is limited: only one figure is shown (in Fig. 2B), and it is focused on promoter regions. However, H3K4me1 is enriched at enhancers. Therefore, it might be wise to analyze enhancer regulation (and integrate H3K4me1 enhancer analysis with additional ChIP-seq data generated in the project, including Kdm1a, H3K27me3, H3K27ac... ChIP-seq). Indeed, loss of clear boundaries between repressed regions containing upDEGs and active domains in which they are imbedded could result, at least in part, from altered regulation of enhancers. Kdm1a has been shown to suppress transcription by silencing enhancers: are H3K4me1/2 increased at enhancers in Kdm1a ko cells, which could contribute to upDEGs?

We thank Reviewer #2 for his/her suggestion. Our manuscript did not include a deeper description of H3K4me1 profiles because the statistical analyses conducted with these samples did not retrieve significantly enriched regions nor significant changes between genotypes. The broad domain distribution of H3K4me1 combined with the limited sample size prevented a more detailed analysis in the case of this histone modification. Still, as suggested by the reviewer, we plotted H3K4me1 signal at the enhancers associated with Kdm1a and did not observe changes in H3K4me1 level when Kdm1a-*if*KO and control samples were compared. This result has been included in the manuscript and it is described in page 9 and shown in the new panel **Supp. Fig. S3E**.

2- What is the overlap/correlation between RNA-seq DEGs and ChIPseq DERs upon Kdm1a neuronal depletion?

We thank the Reviewer for suggesting these new analyses. We found a significant correlation between the changes in H3K4me3 and the changes found in RNA-seq. We also present the negative correlation between H3K27ac and H3K27me3. These results are described in page 9 and shown in the new panels **Supp. Fig. S3D** and **S3F**.

3- *The authors show that promoters of upDEGs are enriched in H3K27me3 and also, to some extent in H3K4me3, which suggest that Kdm1a-sensitive genes are bivalent genes. Could the authors specifically analyze this? What is the proportion of upDEGs which are bivalent, that the presence of H3K4me3 at upDEG correlate with Kdm1a binding?*

We thank the reviewer for this question. Unfortunately, it is difficult to determine if a gene is “*bivalent*”. This term was introduced in stem cells studies investigating homogeneous cell populations. Our ChIP-seq data was produced in hippocampal tissue where cell heterogeneity makes such classification more difficult. In addition, the definition of bivalence can be qualitative (based on peak calling) or quantitative (by defining arbitrary thresholds for H3K4me3 and H3K27me3 levels). Using qualitative criteria, the weak H3K4me3 signal detected at upDEGs in the chromatin of control mice did not reliably detect peaks at these genes leading to the conclusion that upDEGs are only enriched in H2K27me3. Using quantitative criteria, as already shown in **Fig. 1I** and **Fig. 2E**, we detected a low signal for H3K4me3 in the promoter of most upDEGs, but that signal is much lower than the one found in neuronal genes (i.e., genes highly expressed in neurons). The conclusion in both cases would be that we cannot consider that the upDEGs are bivalent genes, at least that the threshold for H3K4me3 signal in the definition is severely reduced. The attached figure shows the levels of H3K4me3 in Kdm1a-ifKOs and control littermates in 3 sets of genes: upDEGs in Kdm1a-ifKOs, genes rich in H3K27me3 that are not affected in Kdm1a-ifKOs and a random set of genes expressed in neurons. There is no difference between the first two sets.

4- *What is the impact of Kdm1a loss in neurons on activity-regulated genes? The authors have treated mice with KA and show that Fos is increased upon KA treatment both in Kdm1a KO mice and control mice. What about additional IEGs, including Egr1, which is strongly dependent on SRF. It might be interesting to investigate such IEGs, since corepressor complex comprising Kdm1a was implicated in the repression of SRF target genes.*

As indicated by the Reviewer, experiments in neuroLSD1^{HET} mice have involved Kdm1a in the repression of SRF targets (Rusconi *et al.*, *PNAS*, 2016). To test if this was also the case in Kdm1a-ifKOs, we used the glutamate agonist kainic acid (KA) that triggers a strong transcriptional response in hippocampal cells, including many SRF targets. However, we could not detect any significant deficits in the induction of IEGs in Kdm1a-

ifKO neurons. RT-qPCR assays in the hippocampus of *Kdm1a*-ifKOs and control littermates showed a similar induction of mRNA levels for *Egr1* and *Arc*, two direct targets of SRF, as well as for *Fos*, downstream of CREB1, 1 h after KA treatment. These results suggest that proper *Kdm1a* levels, particularly of its neuron-specific isoform, are crucial to establish proper levels of induction for SRF target genes during development, but later in the adult brain it may become redundant, likely because other proteins also participate in the precise regulation of IEG inducibility. The new RT-qPCR assays are presented in the new panel **Supp. Fig. S1K**.

5- *What is the expression level of upDEG-associated genes (they are expected to be repressed if H3K27me3 is increased, as shown in Fig.4D)? RNA-seq data using rpkm could be used to show expression results, it is more quantitative than metaprofiles. Additionally, how to explain that generally *Kdm1a* loss has no impact on transcription of active genes? It could be discussed*

As shown in **Fig. 1F**, the expression level of upDEGs in principal neurons is very low. We have now extended this analysis by providing additional information comparing the raw count data for upDEGs in control samples with the expression of all other protein coding genes (new **Supp. Fig. S2A**).

Regarding the second question, “*why are the genes that *Kdm1a* directly binds not affected transcriptionally?*”, we do not have a definite answer. Epigenetic regulation in postmitotic cells, and likely particularly in neurons, is very robust and highly redundant. It is likely that other KDMs collaborate with *Kdm1a* in active genes to maintain proper levels of lysine methylation. Our data indicate that only a relatively small set of genes remain sensitive to *Kdm1a* dose in differentiated neurons. In addition, our data in hippocampal culture (**Supp. Fig. S8L**) support a role for *Kdm1a* beyond its catalytic activity, consistent with recent evidence in other systems (Gu *et al.*, *Cell Mol Life Sci* 2020 and Casey *et al.*, *iScience* 2023). Therefore, it is possible that the main non-redundant function of *Kdm1a* in adult neurons is to preserve the boundaries between active and inactive domains at specific genomic locations, which could explain the limited impact of its loss in the transcriptionally active associated genes. This is now discussed in page 20.

6- *Fig6C-F, analysis of positive and negative control proteins (with and devoid of LLPS properties, respectively) would strengthen the results*

We conducted the experiment suggested by the Reviewer. Unfortunately, as shown in the images below, that expression of truncated *Kdm1a* in HEK cells interfered with mitosis

leading to cells with enlarged nuclei that failed to divide. This finding is consistent with recent studies demonstrating that Kdm1a is required for centrosome duplication and normal chromosome segregation (Wang et al., *Nat Genet.* 2009, 41:125-9; Venugopalan et al., *Mol Cell Biol.* 2012 32:4861–4876; Adamo et al., *Nat Cell Biol* 2011, 13:652–659; Lv et al., *Eur J Cell Biol.* 2010, 89:557-63). Since our study focused on the role of Kdm1a in postmitotic neurons we have not investigated further this result.

7- P17, line 398, “this result is consistent with a recent study showing that the euchromatinization of H3K27me3-repressed genes is a key feature of aging tissues (Yang et al. 2023b)”. In this paper focusing on mouse liver, the authors show that aging is associated with both derepression of PRC2 target genes (as found in the current study) and global increase of H3K27me3 at age-domains/LADs, which are broad gene-poor domains. It would be interesting to examine whether loss of Kdm1a also leads to epigenetic changes at those broad domains, which would provide further evidence that Kdm1a loss mimicks aging.

We thank the Reviewer for his/her suggestion. Unfortunately, we did not conduct ChIP-seq experiments for H3K27me3 in elderly Kdm1a-*if*KO mice. Also, we cannot conclude that there is a global increase in H3K27me3 in age-domains/LADs based on our ChIP-seq data. The datasets generated by Yang et al., 2023b revealed this effect because they used spike-in in their ChIP-seq experiments. This made possible to report a global change in H3K27me3 that otherwise would not have been detected.

Minor

-Fig.1D, other histone demethylases than Kdm1a or histone methyltransferase are not changed upon Kdm1a inactivation, as shown by RNA-seq analysis. However, this does not exclude changes at protein level/activity. The conclusion should be tone down/discussed

We revised the indicated sentences in page 6. The new text reads: “The analysis also demonstrated the absence of compensatory upregulation of other genes encoding KDMs or the downregulation of lysine methyltransferases (KMT)-encoding genes, at least at the transcriptional level”

-Fig2E comes before Fig2D, this needs to be corrected

We thank the Reviewer for detecting this mistake. We revised the text to make sure that all the panels are cited in order.

-FigS3E: Cbx1,3 and 5 are not PRC1 subunits; Cbx2/4/6/7/8 are PRC1 subunits, their profiles should be shown instead of that of Cbx1/3/5.

We apologize for this mistake. We corrected the text and incorporated the new profiles in **Supp. Fig. S3H** (former **Supp. Fig. S3E**).

-The two graphs in Fig3A are unclear, the legend is not enough explicit (what is 8B3.3, 4D3...?).

The graph in **Fig. 3A** shows the enrichment analysis of upDEG at genomic locations conducted at *Webgestalt*. The analysis revealed that upDEGs are enriched at specific cytogenetic bands. We have revised the legend of **Fig. 3A** to explain the abbreviations in more detail (page 35).

-In Fig3B, the difference does not appear huge, what is the statistics?

We used the Kolmogorov-Smirnov test, $D = 0.14$, $p = 0.000042$. The significance is indicated in the legend of **Fig. 3B** in page 35.

-Fig. 3D aims to show that upDEGs are organized in clusters. Could this be quantified, relative to gene density for instance?

We are not sure about how to perform the analysis requested by the Reviewer. We do not believe that gene density is different in the genomic regions that contain upDEGs. In fact, there is no reason to believe that. Our observation refers to an enrichment for upDEGs at specific cytogenetic bands shown in **Fig. 3A**. Also, to complement this finding, we conducted the proximity analysis presented in **Fig. 3B**.

Fig.3G, ChiAPET data. The legend of Fig3G could be more detailed, minimal information is provided.

We extended the legend of **Fig. 3G** in page 35. We hope that the Reviewer will find the addition of new information satisfactory.

-Fig3H, how is loop span calculated?

In **Fig. 3H**, “CTCF pet counts” indicate the addition of ChIA-PET counts at CTCF loops that encapsulate 100% of the gene body (the longest loop if there are multiple isoforms). We did not impose any rules on distance from the gene. To calculate the loop-span in the right panel of **Fig. 3H**, the same encapsulation loops were taken and the average distance between the CTCF peaks that encapsulate the genes was measured. The revised version of the manuscript includes this information in the legend of **Fig. 3H** and explain in more detail the elaboration of these graphs in page 11 of *Experimental Procedures*.

-FigS4D, what is the distribution for all genes? To suggest that upDEGs are enriched in CTCF regulated genes, a reference for comparison is needed.

We apologize if we were not clear in our conclusions. We believe that the Reviewer may have misinterpreted us. We do not indicate that upDEGs are enriched in CTCF regulated genes. Indeed, this is not the case. The percentage of upDEGs associated with CTCF peaks (62% as shown in former **Fig. S4D**) is very similar to the one obtained if we conduct the same analysis with a list of random neuronal genes (59% of the genes expressed in neurons are associated with CTCF peaks using the same criteria). The notion that we were attempting to convey was different. We hypothesized that upDEGs display greater topological control compared with other genes because of the enrichment analysis shown in **Fig. 1H** and the loop analysis presented in **Fig. 3H**, which indicates greater CTCF-dependent topological control in upDEGs genes compared to unchanged non-neuronal genes and neuronal genes.

-The statistics shown in FigS4F are unclear

In this and other figures, we calculated the statistical significance of the overlap between two groups of genes with the online tool available at nemates.org. According to this test, a representation factor > 1 indicates more overlap than expected by chance, while a representation factor < 1 indicates less overlap than expected by chance. The accompanying statistic is based on exact hypergeometric probability. The revised version of *Experimental Procedures* includes this information (page 13 of *Supplementary Materials*).

-Fig.4C-D, is it possible to quantify the proportion of upDEGs associated with loops, presenting one anchor in an active region bound by Kdm1a? and p12 & FigS6A: “some CTCF loops presented H3K27me3 at one CTCF anchor and H3K27ac and KDM1a at the opposite” Could it be quantified? What about the other loops with upDEGs?

This point is important, and we tried to address it using our ChiA-PET data. It is however difficult to quantify this properly because several CTCF loops may come together in the proximity of a single upDEG. As shown in the examples in **Fig. 4E** and **Supp. Fig. S6D**, the genes *Spint1* and *Aap5* have multiple CTCF connections, both upstream and downstream of the gene. The analysis of primary loops indicated that at least 48% of the uDEGs show this pattern but the number will be larger if secondary loops are also considered. Due to this complexity, we believe that it is not possible to provide a specific number.

-In Fig.S6BC, ChIPseq profiles in both WT and Kdm1a ko should be shown when possible

We agree with Reviewer #2 and tried to include this information whenever possible. We revised **Supp. Fig. S6C** to include the epigenetic profiles for control and *Kdm1a*-ifKO samples. However, we did not change **Supp. Fig. S6B** because we thought that this figure was already too large and complex to add additional information, particularly considering that the differences between genotypes is already shown in **Supp. Fig. S6D**.

-Fig.4D, it seems that H3K27me3 is increased at CTCF peak associated with up DEG-associated genes, and that H3K4me3 and transcription are also low (see ‘iv’) Also strikingly RNA peaks (see ‘iv’) are greater at TES than TSS, why is it so?

Regarding the first question, we apologize for the wrong labeling of some panels in **Fig. 4D**. The panels in columns (i) and (iv) should indicate *Center* (of ChIP-seq peak) instead of *TSS*. We believe our mistake may have confused Reviewer #2. It is not surprising that H3K4me3 and transcription are low in these locations (they are not TSSs). We corrected this mistake in the revised figure.

Regarding the second question, we are not sure we understand it. Maybe the Reviewer was referring to RNA peaks in panel (iii) instead of panel (iv). As expected, the signal for mRNA in panel (iii) is greater at TES than TSS because the mRNA-seq libraries were made using oligo(T) primer and this produces an enrichment for the 3' sequence of the transcripts.

-Fig4D. the representation (i), (ii)....in this figure is unclear: the CTCF peak directly linked to upDEGs is centered to TSS (see i) and it seems it does not match with the TSS of upDEGs shown in (ii). Similarly, the CTCF peak directly linked to upDEG-associated

genes is centered to TSS (iii), but it does not seem to match to what is shown in (iv). Peak intensities are different. It is unclear why CTCF peaks are centered to TSS.

As indicated in the previous point, we apologize for the wrong labeling of some panels in **Fig. 4D**. We believe that this mistake confused Reviewer #2. The CTCF signal that was plotted in the panels in columns (i) and (iv) does not correspond to the TSS but to the center of the CTCF peak. We corrected this mistake in the revised figure.

-Fig.4E, it would be wise to also show mRNA levels of Spint1, Vps18, Ino80 and additional genes in the region, in Kdm1a ko and control conditions. Also H3K4me3 and H3K4me1 profiles could be included. Generally, profiles of both control and Kdm1 ko could be shown.

Although we agree with Reviewer #2 that more information is usually useful, we believe that adding the RNA-seq and Chip-seq tracks for control and Kdm1a-ifKO samples to a figure that already contains abundant information would be confusing. In any case, we show below a revised version of **Fig. 4F** that includes the requested information. We could replace the original panel by this panel if the Reviewer consider that this would improve the paper.

Alternative **Fig. 4F** including the profiles for H3K27me3 and H3K27ac ChIP-seq; and RNAseq from the hippocampal samples of Kdm1a-ifKOs and control littermates.

-Fig.4F and S6G (4Cseq data): changes in interaction strength in Kdm1a vs control at Spint1 is unclear. The representation/figure legend could be improved. Statistics for 4Cseq data are not shown, though it is said that differences are significant. It is difficult to know the extent of the changes. A representation showing 4Cseq signals would be more informative.

Moreover, data related to Ppp1r14d are not shown, only data for Spint1 are shown.

We have included in the revised **Supp. Fig. S6** the mean signal for 4C-seq in control mice and Kdm1a-ifKOs. This new panel (**S6H**) shows the changes detected using DESeq2. The statistic for the differential analysis is presented in a new Supplementary Table (**Table S3**).

Regarding the last point, *Ppp1r14d* is adjacent to *Spint1*. The 4C-seq was performed using the *Spint1* promoter as anchor. We revised the text to clarify this information (page 13 of the main text and page 27 of *Supplementary Materials*).

-Generally, legend figures could be improved, particularly providing more details.

We revised the figure legends to improve the readability of the manuscript.

Reviewer #3 (Remarks to the Author):

*The manuscript by del Blanco et al. presents a nice and rather thorough investigation of the role of *Lsd1* (*Kdm1a*) in neurons of the adult mouse forebrain, based on epigenomic, transcriptomic, and genomic architectural analysis of conditional knockout mice. They also deftly utilize publicly available multi-omics data to support their model, which proposes that *Lsd1*, by virtue of its N-terminal IDR, maintains the repression of nonneuronal genes in neurons through direct and indirect collaboration with the PRC2 complex and CTCF, respectively.*

*As the authors note, a few studies have already addressed the potential function of *Lsd1* in neurons; however, the manuscript does not sufficiently integrate their new results with previously published data on this topic. At the very least, additional explanation is needed to address any discrepancies as well as the relevance of unexplored aspects, some of which are highlighted below:*

We thank Reviewer #3 for his/her comments. We extended the Discussion of our results to highlight key differences between our study and previous studies (pages 18-19). Specifically, we indicate that Laurent and colleagues investigated the role of the neuronal *Kdm1a* isoform (n*Kdm1a*) during neuronal differentiation (Laurent *et al.*, 2015). Likewise, other relevant studies investigated the role of n*Kdm1a* using conventional knockout mouse strains (Rusconi *et al.*, 2016) or early gene ablation using a Nestin-Cre driver (Wang *et al.*, 2015). Therefore, these studies focused on neuronal development and could not dissect the developmental and adult functions of *Kdm1a*. In our study, we aimed to specifically investigate the role of *Kdm1a* in mature neurons of the adult mouse brain, eliminating the confounding effect of *Kdm1a* ablation at earlier stages of neuronal differentiation. To do this, we specifically eliminate *Kdm1a* in 2-month old mice or older using the *Camk2-Cre-ERT2* tamoxifen-inducible recombination system. The study of Christopher *et al.* (now discussed in greater extent) also used a similar inducible strategy but the promoter driving the expression of the Cre recombinase (*Cagg-CreERT2*) is ubiquitously active, therefore gene ablation takes place in most (if not all) cell types and organs.

*• The difference in phenotype and hippocampal gene expression changes in the conditional *Lsd1* knockout mice in this study vs. earlier work (Christopher et al., 2017) that employed a *Cagg-Cre* tamoxifen-inducible system is striking and rather surprising. Is there any overlap in the RNA-seq changes detected in the two studies?*

Most of the genes upregulated in Christopher *et al.*, likely reflect the severe phenotype observed in these mice, including neurodegeneration, gliosis and inflammation. As discussed in pages 5 and 19, we believe that the differences between the two studies rely

on the specificity of gene ablation: *Kdm1a* is specifically lost in the adult excitatory forebrain neuron in our study, but ubiquitously lost in Christopher et al. Still, we found an overlay of 30 genes between both differential gene expression analyses, including *Spint1*, which suggest that the gene de-repression signature can be also detected in these mice, although largely occluded by the inflammation response.

• Two different neuronal isoforms of Lsd1 have been described, each with altered histone specificity (Laurent et al., 2015; Wang et al, 2015). These isoforms are presumably absent in the conditional knockout. Accordingly, are any global changes in H4K20me1/2 or H3K9me2 expected that might contribute to the observed gene expression changes?

We cannot discard this option. Our work focuses on the study of H3K4 methylation balance, which has previously been identified as a direct target of *Kdm1a* in cooperation with the CoREST complex, and the impact on H3K27 methylation/acetylation given the association of upDEGs with PRC2. We, however, did not investigate H3K9me2, H4K20me1/2 or other histone modifications because such exhaustive analysis of hPTMs changes was out of the scope of our study. Still, to answer Reviewer #3 we conducted a basic screen for alterations in H3K9me2 in *Kdm1a*-ifKOs using immunostaining. Confocal microscopy images with antibodies that recognize H3K9me2 do not show differences in distribution and signal intensity between *Kdm1a*-ifKO mice and control littermates. We should also note that we did not observe transcriptional changes in the induction of IEGs in response to kainic acid (KA) administration

(Supp. Fig. S1L and new panel S1K in response to Reviewer #2), which could suggest that H4K20me1/2 is not altered in *Kdm1a*-ifKOs since changes in this histone mark have been correlated with transcriptional elongation after neuronal activity (Wang et al, 2015).

Images show H3K9me2 staining in CA1 neurons of *Kdm1a*-ifKOs and control mice

It is interesting that few of the de-repressed genes in the conditional Lsd1 knockout neurons actually recruit Lsd1. The authors note that ~60% of these dysregulated genes harbor proximal CTCF sites, but what do the authors think is going on with the remaining ~40%? Do these have more distal CTCF sites that might still be functionally significant, or do the authors suspect that other collaborating factors may be involved?

As indicated by Reviewer #3, our enrichment analysis was limited to CTCF peaks located up to 10 Kb upstream of the TSS and 10 Kb downstream from the TTS of upDEGs. This

criterion only defines the lower limit for the association between upDEGs and CTCF binding. We cannot discard that more distal CTCF binding could also play a topological function. In addition, we cannot rule out that other molecular mechanisms are involved in the de-repression of genes that do not show CTCF binding.

This sentence in the Supplementary Figure S8 caption is confusing and seemingly inaccurate: “RT-qPCR assays demonstrate that human Kdm1a bearing mutations (K661A and K598E) that abolish its H3K4me2 demethylase catalytic activity do not prevent upDEG activation as efficiently as the hKdm1a protein lacking the IDR.” The LSD1 catalytic mutants largely prevent de-repression, unlike the IDR deletion mutant, which is presumably the point.

We thank the Reviewer for pointing out our mistake. We have corrected this sentence in the legend of **Supp. Fig. S8L**. The revised text is “*RT-qPCR assays demonstrate that human Kdm1a bearing mutations that affect H3K4me2 catalytic activity (K661A and K598E) largely prevented the activation of upDEGs, although less efficiently than hKdm1a.*”

Finally, regarding the suggestion that de-repression of Lsd1- regulated genes might be a feature of brain aging, how do the authors envision the involvement of Lsd1 itself? Is there any evidence that levels of Lsd1 (or of PRC2 components) change in neurons with age?

Kdm1a has recently been associated with neurodegenerative disorders such as Alzheimer's disease and frontotemporal dementia. In particular, Kdm1a was associated with pathological Tau aggregates in the brain of AD patients, resulting in a reduction of Kdm1a in the neuronal nucleus and mislocalization in the cytoplasm (Christopher et al., 2017). On the other hand, recent studies show that during natural aging there is an increase in protein aggregates at the cytoplasm of brain cells (Cuanalo-Contreras et al., *Front Aging Neurosci* 2022). These findings could suggest that Kdm1a can be trapped in aging-related aggregates and reduce the effective presence of Kdm1a in neuronal chromatin. Our experiments, however, did not reveal a significant increase of cytoplasmic Kdm1a levels or reduction of nuclear Kdm1a in principal neurons of 20-month-old mice. This information was added to page 23 in the *Discussion* section.

These outstanding questions/issues notwithstanding, the manuscript of del Blanco et al. is instructive and should add an important increment to our understanding of the functional significance of LSD1 in mature neurons

We thank again Reviewer #3 for his/her suggestions, positive appreciation, and useful comments.

REVIEWER COMMENTS

Reviewer #1 (Remarks to the Author):

The authors responded to each comment and properly revised the manuscript. I do not have additional comments and congratulate them on this excellent study.

Reviewer #2 (Remarks to the Author):

The authors nicely addressed the questions raised by the reviewer.

Reviewer #3 (Remarks to the Author):

Most of my comments/concerns have been adequately addressed in the revised manuscript. However, I am still fixated on the finding that few of the de-repressed genes in the conditional Lsd1 knockout neurons actually recruit Lsd1. Has a similar discordance in Lsd1 binding and its regulated target genes been observed in other published studies? There now seem to be many such datasets ascertained from various cell types or treatment conditions. Is it possible that the ChIP-seq distribution observed here is more a consequence (or artifact?) of the experimental procedure rather than the actual biology?

Point-by-point response to the reviewers' comments

We thank Reviewer 1 and 2 for the positive evaluation of the revised manuscript.

Regarding the comment of Reviewer 3, we respond the following:

Most of my comments/concerns have been adequately addressed in the revised manuscript.

We thank the reviewer for appreciating our effort to address all the comments and concerns.

However, I am still fixated on the finding that few of the de-repressed genes in the conditional Lsd1 knockout neurons actually recruit Lsd1. Has a similar discordance in Lsd1 binding and its regulated target genes been observed in other published studies? There now seem to be many such datasets ascertained from various cell types or treatment conditions.

We were also initially surprised by the relatively low percentage of direct binding events involving Kdm1a at the promoters of Differentially Expressed Genes (DEGs). However, such disparities between the binding patterns of epigenetic regulators and their corresponding transcriptional effects are not uncommon in large-scale genomic studies. The overlap between these two datasets is often only partial and not all-encompassing. DEGs frequently do not exhibit direct binding by the regulators of their expression. This can be attributed to several factors, such as long-range interactions that may escape conventional proximal binding analysis, or the influence being mediated indirectly through as-yet-undetermined mechanisms. In the context of Kdm1a's binding profile in principal neurons, our findings strongly suggest that long-range interactions play a pivotal role, as illustrated by the evidence presented in Figure 4 and Supplementary Figures S5 and S6. Nevertheless, we cannot rule out the possibility that some of the changes identified in the RNA-seq analysis may be the result of indirect mechanisms that are neither transcriptional nor epigenetic in nature. Previous studies on Kdm1a KO exploring the correlation between Kdm1a binding and transcriptional changes focused on developmental stages in which the transcriptional impact seems to be much broader and did not produce comparable results. As indicated in page 22 in our manuscript, the importance of Kdm1a regulating gene repression decreases at later stages when repressed genes acquire a permanent silent status by the action of alternative epigenetic mechanisms.

Is it possible that the ChIP-seq distribution observed here is more a consequence (or artifact?) of the experimental procedure rather than the actual biology?

We do not believe that the ChIP-seq distribution is a consequence of the experimental procedure for several reasons:

1. The antibody against-Kdm1a used in the ChIP-seq experiment (Abcam Cat# ab17721) recognizes Kdm1a with great specificity and sensitivity and do not show unspecific binding according to immunohistology experiments. This can be clearly seen in Figure 1c where the signal for Kdm1a detected in principal neurons completely disappeared in ifKO mice, as well as in Figure 5a (using super-resolution microscopy) and Supplementary Figure S1a-d.

- As described in lines 277-279 and shown in Figure 4e and Supplementary Figure S5a, the profile for Kdm1a binding in hippocampal chromatin of adult mice generated in the frame of this study is very similar to that published by Mukai and colleagues in adult mouse prefrontal cortex (Mukai et al., *Neuron*, 2019). The same rabbit polyclonal antibody against hKDM1A (Abcam Cat# ab17721) was used in both ChIP-seq experiments. It is unlikely that two independent laboratories generate the same artifactual result.
- A third study exploring Kdm1a binding to neuronal chromatin, this time using a different antibody, also revealed a similar occupancy profile. In this study (Wang et al., *Nat Neurosci*, 2015), the authors overexpressed Flag-neuronal-Kdm1a or Flag-constitutive-Kdm1a in primary neuronal cultures derived from Kdm1a-KO mice and used an anti-Flag antibody to generate the binding profile for these two protein isoforms. Furthermore, they compared the binding profile of the two flag-tagged proteins in cultures that were either exposed to or not exposed to potassium chloride (KCl). The ChIP-seq profiles obtained from the cultured neurons closely resembled those obtained from the adult hippocampus (see Figure below). We added this information to **Supp. Fig. S5** in our manuscript. One of the key conclusions drawn from this study was that, much like our own findings, Kdm1a predominantly associates with active promoters and enhancers of actively transcribed genes. This same binding pattern was also observed in the seminal investigation of Kdm1a's occupancy profile in embryonic stem cells (Whyte et al., *Nature*, 2012).

The enclosed figure provides genomic snapshots encompassing two upregulated differentially expressed genes (upDEGs) discussed in our study: *Tmc8* and *Kcna7*. The comparison of Kdm1a ChIP-seq profiles generated in three independent studies revealed enrichment in the same regions. In alignment with our findings (del Blanco et al., 2023), the profiles from cortex (Mukai et al., 2019) and neuronal cultures (Wang et al., 2015) also indicate that Kdm1a does not directly bind to the upDEGs.

REVIEWERS' COMMENTS

Reviewer #3 (Remarks to the Author):

The authors have adequately addressed my comments/concerns. In my opinion, this interesting study is suitable for publication in Nat Commun.